# Leaving No OOD Instance Behind: Instance-Level OOD Fine-Tuning for Anomaly Segmentation

**Yuxuan Zhang[1], Zhenbo Shi[1,2*], Han Ye[1], Shuchang Wang[1], Zhidong Yu[1,2], Shaowei Wang[4], Wei Yang[1,2,3*]**

[1]School of Computer Science and Technology, University of Science and Technology of China
[2]Suzhou Institute for Advanced Research, University of Science and Technology of China
[3]Hefei National Laboratory, University of Science and Technology of China
[4]Institute of Artificial Intelligence and Blockchain, Guangzhou University

## Abstract

Out-of-distribution (OOD) fine-tuning has emerged as a promising approach for anomaly segmentation. Current OOD fine-tuning strategies typically employ global-level objectives, aiming to guide segmentation models to accurately predict a large number of anomaly pixels. However, these strategies often perform poorly on small anomalies. To address this issue, we propose an instance-level OOD fine-tuning framework, dubbed LNOIB (**L**eaving **N**o **O**OD **I**nstance **B**ehind). We start by theoretically analyzing why global-level objectives fail to segment small anomalies. Building on this analysis, we introduce a simple yet effective instance-level objective. Moreover, we propose a feature separation objective to explicitly constrain the representations of anomalies, which are prone to be smoothed by their in-distribution (ID) surroundings. LNOIB integrates these objectives to enhance the segmentation of small anomalies and serves as a paradigm adaptable to existing OOD fine-tuning strategies, without introducing additional inference cost. Experimental results show that integrating LNOIB into various OOD fine-tuning strategies yields significant improvements, particularly in component-level results, highlighting its strength in comprehensive anomaly segmentation.

## 1 Introduction

Semantic segmentation has achieved remarkable success in autonomous driving. However, traditional segmentation approaches adhere to a closed-set training taxonomy [52, 44, 55]. When deployed in real-world scenarios, these segmentation networks struggle to predict instances of previously unseen categories (known as OOD instances or anomalies), inevitably leading to potential risks.

To identify OOD regions, great efforts have been made in anomaly segmentation (AS) [21, 48, 31, 62, 57, 35]. One promising solution is OOD fine-tuning [4, 47, 41], which uses mixed-content images (containing ID and OOD regions) to enhance the generalization of segmentation models to previously unseen instances. Current OOD fine-tuning strategies typically use global-level objectives, aiming to accurately predict as many anomaly pixels as possible, rather than segmenting all anomalies. As a result, these objectives lead to the neglect of small OOD instances. However, mispredicting small anomalies (e.g., a small cow in Figure 1) can pose significant safety risks. Thus we argue that detecting small anomalies is also critical for ensuring the safety of segmentation networks.

In this paper, we propose a novel OOD fine-tuning framework called LNOIB to ensure a comprehensive detection of all OOD instances by the segmentation model. We start by theoretically analyzing

---

*Zhenbo Shi and Wei Yang are corresponding authors. E-mail: {zbshi,qubit}@ustc.edu.cn

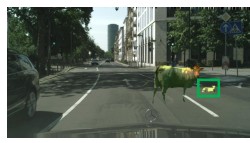 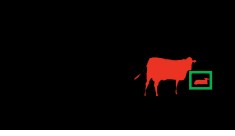 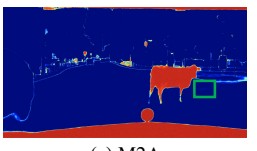 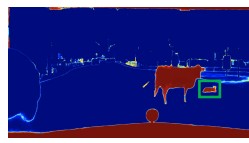

(a) input image    (b) OOD ground-truth    (c) M2A    (d) M2A+LNOIB (ours)

Figure 1: Current OOD fine-tuning strategies for AS, such as Mask2Anomaly (M2A) in (c), sometimes neglect small anomalies (e.g., a small cow in the green box). When incorporating LNOIB, M2A effectively segments both large and small anomalies (e.g., both big and small cows in (d)).

why current global-level objectives for OOD fine-tuning often fail to detect small anomalies. For this reason, we introduce a simple yet effective instance-level objective that equally optimizes each anomaly. Note that, beyond a specific loss function, this objective serves as a versatile function that can be adapted to various existing global-level OOD fine-tuning losses (such as entropy-based loss [4], energy-based loss [47], etc.). By integrating both global-level objective and our instance-level objective, we formulate an overall *prediction-based objective*, which effectively guides the model to detect small anomalies while maintaining high accuracy in identifying large anomalies.

Moreover, another reason for the neglect of small anomalies is that the features of OOD instances often become smoothed by their ID surroundings during convolution operations [54, 46]. To tackle this issue, we propose a *feature separation objective* to ensure that the prototype of each OOD instance diverges significantly from ID representations. This objective comprises an ID semantic loss and a nearest neighbor loss: the former provides a universal view to prevent OOD features from aligning with ID semantic prototypes, while the latter ensures that each OOD prototype remains distinct from its nearest ID prototype neighbors.

LNOIB combines the prediction-based objective with the feature separation objective to effectively segment all anomalies. Accordingly, LNOIB offers several advantages to OOD fine-tuning: 1) Effectiveness: The proposed objectives in LNOIB significantly enhance the overall performance, particularly in small anomalies. 2) Efficiency: LNOIB incurs no additional computational cost during inference, ensuring a consistent inference speed. 3) Versatility: LNOIB can be seamlessly adapted to existing OOD fine-tuning strategies, such as Entropy Maximization (EM) [4], PEBAL [47], and M2A [41]. Note that LNOIB acts as a paradigm that can be integrated into existing OOD fine-tuning strategies, rather than being limited to a specific loss function. Experimental results show that extended by LNOIB, current OOD fine-tuning strategies achieve better performance across various benchmarks, with notable improvements in component-level and instance-level metrics.

We briefly summarize our contributions as follows:

- We propose LNOIB, a novel OOD fine-tuning framework for AS, which adapts existing strategies that merely use global-level objectives to improve the segmentation results of small anomalies.
- We introduce a prediction-based objective and a feature separation objective within LNOIB to guide the segmentation models in focusing on small OOD instances.
- Experimental results show that integrating LNOIB significantly enhances the performance of existing OOD fine-tuning strategies (EM, PEBAL, and M2A), particularly in component-level and instance-level metrics.

## 2    Related Work

**Anomaly Segmentation** AS aims to segment instances whose categories are not present in the training dataset. Existing approaches can be broadly categorized into discriminative methods [21, 34, 24, 17, 58] and generative methods [16, 48, 10, 27, 29, 18]. The former primarily utilizes the predictions of semantic segmentation to estimate uncertainty, while the latter typically employs extra generative networks to model the distribution of ID samples. Both methods produce a pixel-wise anomaly score map as the result of AS. Accordingly, the most commonly used pixel-level metrics measure the accuracy of pixel predictions. However, this focus can lead to the neglect of small OOD instances, as they have less impact on the overall performance. In this paper, we mainly compare component-level and instance-level metrics as introduced in [3, 37] to evaluate LNOIB.

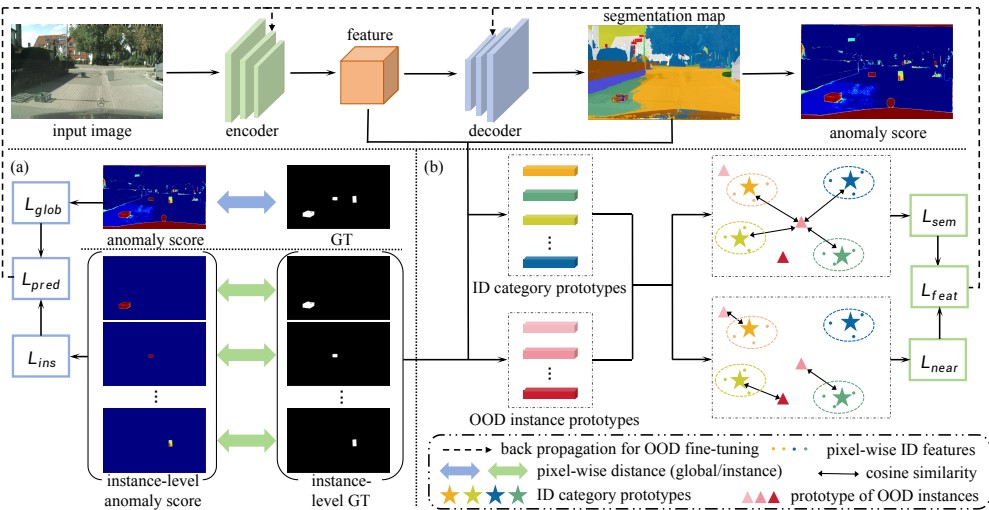

Figure 2: Overview of LNOIB fine-tuning framework for AS. (a) illustrates the prediction-based objective containing a commonly-used global-level objective $L_{glob}$ and a novel instance-level objective $L_{ins}$; (b) depicts the feature separation objective, including an ID semantic loss $L_{sem}$ (with some connections omitted for clarity) and a nearest neighbor loss $L_{near}$.

**OOD Fine-Tuning** OOD fine-tuning is an optional yet effective strategy in discriminative methods for AS. The main goal is to use a mixed-content dataset (containing both ID and OOD regions in each image) to simulate anomalies, thereby exposing the models to outliers and enabling them to learn the differences between ID and OOD patterns [22, 40, 38]. Common approaches for generating these mixed-content datasets follow a direct cut-and-paste strategy [47, 31]. Moreover, a suitable fine-tuning objective is required to guide the model in distinguishing between ID and OOD data. Existing literature often employs softmax entropy [4], energy [47], or logit [36] to design fine-tuning objectives in a pixel-wise manner. However, these objectives tend to focus on correctly predicting more pixels globally, falling short of detecting small anomalies. In this paper, we propose a novel OOD fine-tuning framework, termed LNOIB, to segment both large and small anomalies.

**Prototype Learning** Prototypes can be regarded as the mean representations of semantic category features in few-shot segmentation [50, 25, 26, 60, 33]. Existing approaches typically utilize category-wise prototypes to align pixel-wise features and produce dense predictions [50, 26]. In contrast to previous studies, this work investigates whether prototype-based representations can benefit the OOD fine-tuning process. Specifically, we construct ID prototypes for each ID category and OOD prototypes for each OOD instance, and encourage clear separation between OOD and ID prototypes. The motivation for adopting instance-wise OOD prototypes, rather than merely enforcing the separation between ID category prototypes and all pixel-wise OOD representations, is also supported by our instance-level theoretical analysis to perform well on small anomalies.

## 3 Methodology

We begin with a brief review of AS. Then we theoretically analyze the limitations of existing global-level fine-tuning objectives and introduce a versatile instance-level objective. Next, we propose a feature separation objective to enhance the detection of small anomalies. Finally, we present the overall OOD fine-tuning framework of LNOIB.

### 3.1 Task Preview

Given an input image $x \in \mathbb{R}^{3 \times H \times W}$, a segmentation model is utilized to extract the latent features $F \in \mathbb{R}^{C \times H \times W}$ of $x$ through the encoder, where $C$ is the feature dimension. The decoder of the model then processes these features to produce the segmentation results $f(x) \in \mathbb{R}^{K \times H \times W}$, where $K$ is the total number of ID categories.

To segment anomalies, AS yields a pixel-wise anomaly score map $s(x) \in [0,1]^{H \times W}$ based on $f(x)$. For example, the pioneering work [21] uses the maximum softmax probability of $f(x)$ as the confidence score to produce $s(x)$. However, segmentation models often exhibit overconfidence in their predictions. To address this, OOD fine-tuning is a widely adopted way to enhance the model, enabling it to separate ID and OOD patterns to refine AS performance. Despite this, current OOD fine-tuning strategies struggle to capture small anomalies. We theoretically analyze the reasons for this limitation below.

### 3.2 Global vs. Instance-Level Objective

Before introducing our approach, we first analyze why existing OOD fine-tuning strategies neglect small anomalies. Given a mixed-content image $x$ and its ground-truth binary mask $y \in \{0,1\}^{H \times W}$ (indicating normality or anomaly), previous OOD fine-tuning strategies typically employ global-level contrastive losses to separate ID and OOD regions, aiming to maximize the accuracy of pixel predictions across the entire image. These global-level objectives can be uniformly formulated as:

$$L_{glob} = \underbrace{\frac{1}{N_{in}} \sum_{x_i \in \Omega_{in}} l_{in}[s(x_i), y_i]}_{L_{in}} + \underbrace{\frac{1}{N_{out}} \sum_{x_j \in \Omega_{out}} l_{out}[s(x_j), y_j]}_{L_{out}} \tag{1}$$

where $\Omega_{in}$ and $\Omega_{out}$ denote the image lattices of ID and OOD regions (based on the binary mask $y$), respectively. $i$ and $j$ are the spatial indices. $N_{in} = |\Omega_{in}|$ and $N_{out} = |\Omega_{out}|$ separately are the total number of pixels in $\Omega_{in}$ and $\Omega_{out}$. Additionally, $l_{in}$ and $l_{out}$ signify the pixel-wise loss for ID and OOD pixels, respectively. Note that, $l_{in}$ and $l_{out}$ can be any metric that measures the similarity between the pixel-wise anomaly score and its ground-truth, including entropy-based loss [4], energy-based loss [47], and logit-based loss [41]. For convenience, we denote the first and second terms of Eq. (1) as $L_{in}$ and $L_{out}$, respectively.

As to the OOD region $\Omega_{out}$, it typically contains several independent OOD instances $O = \{o_1, o_2, ..., o_N\}$, where $N$ is the number of anomalies. Accordingly, we have $\Omega_{out} = \Omega_{o_1} \cup \Omega_{o_2} \cup ... \cup \Omega_{o_N}$. Below, we present Lemma 1.

**Lemma 1.** *(Weighted Loss Decomposition). The total loss $L_{out}$ decomposes into instance-specific components:*

$$L_{out} = \frac{1}{|\Omega_{out}|} \sum_{o_k \in O} |\Omega_{o_k}| \cdot \underbrace{\mathbb{E}_{x_j \sim \Omega_{o_k}} [l_{out}[s(x_j), y_j]]}_{L_{o_k}} \triangleq \sum_{k=1}^{N} w_k L_{o_k} \tag{2}$$

*where $w_k \triangleq \frac{|\Omega_{o_k}|}{|\Omega_{out}|}$ is the normalized weight of instance $o_k$ and $L_{o_k}$ is the mean loss over $\Omega_{o_k}$.*

The proof of Lemma 1 is provided in Appendix A. Based on this decomposition, the overall loss $L_{out}$ can be viewed as a weighted sum of anomaly-wise losses $L_{o_k}$, where the weights correspond to the relative size of each anomaly region. This formulation enables further analysis of the influence of the dominant anomaly on the total loss. That is, when a single OOD instance occupies a large portion of the anomaly region, the following theorem provides a bound on $L_{out}$ in terms of the loss contributed by that dominant OOD instance:

**Theorem 1.** *(Dominant Instance Effect). If there exists $t \in \{1, ..., N\}$, such that:*

$$|\Omega_t| \geq (1 - \epsilon)|\Omega_{out}| \quad \text{for some } \epsilon \in (0, 1), \tag{3}$$

*then the loss $L_{out}$ is bounded by:*

$$(1 - \epsilon)L_{o_t} \leq L_{out} \leq (1 - \epsilon)L_{o_t} + \epsilon \cdot \max_{k \neq t} L_{o_k} \tag{4}$$

The proof of Theorem 1 is given in Appendix A. Based on this, we derive the following corollary:

**Corollary 1.** *(Asymptotic Dominance). If $\frac{|\Omega_{o_t}|}{|\Omega_{out}|} \to 1$ (i.e., $\epsilon \to 0$), then $L_{out} \to L_{o_t}$ in probability.*

*Proof.* From Theorem 1, as $\epsilon \to 0$, both bounds converge to $L_{o_t}$. $\qquad\square$

According to Corollary 1, we observe that small anomalies contribute minimally to $L_{out}$, leading to limited optimization during OOD fine-tuning. This likely explains why small anomalies are often ignored in the commonly used global-level objectives. To address this issue, we propose a simple yet effective instance-level objective that focuses on each separated OOD instance, formulated as:

$$L_{ins} = L_{in} + \frac{1}{N} \sum_{o_k \in O} \sum_{x_j \in \Omega_{o_k}} \frac{1}{|\Omega_{o_k}|} \cdot l_{out}[s(x_j), y_j] \tag{5}$$

where $L_{in}$ is equal to that of Eq. (1). This instance-level objective addresses the above drawback by encouraging segmentation models to equally focus on each anomaly. However, merely employing $L_{ins}$ is insufficient because ensuring the segmentation quality of large anomalies is also important. Specifically, $L_{ins}$ draws attention to all potential anomalies while reducing the penalty on large ones compared to $L_{glob}$. Hence, we incorporate the commonly used global-level loss to $L_{ins}$, and formulate the total prediction-based loss as:

$$L_{pred} = \alpha L_{glob} + (1 - \alpha) L_{ins} \tag{6}$$

where $\alpha$ is a balanced factor that controls the trade-off between global-level quality and instance-level completeness. Note that, $L_{pred}$ is not limited to one specific loss; instead, it acts as a paradigm for existing global-level objectives that calculate the similarity between $s(x_j)$ and $y_j$. For example, if we select the entropy-based function [4] for $l_{in}$ and $l_{out}$ to calculate $L_{glob}$, our proposed $L_{ins}$ will also use the same entropy-based function. This paradigm can be extended to other global-level objectives as well. In this paper, we use the global-level objectives in EM [4], PEBAL [47], and M2A [41] to evaluate the versatility of our proposed instance-level objective. For detailed formulations, please refer to Appendix B.

### 3.3 Features Separation Objective

In addition to using global-level objectives for OOD fine-tuning, another reason for inadequate performance on small anomalies is that the features tend to be smoothed out by their ID surroundings under convolution operations [54, 46]. Therefore, we explicitly constrain the anomaly features to make them more distinguishable to address this issue.

Given an image $x$ and its latent feature $F$, we first calculate the prototype $p_{o_k}$ for each OOD instance $o_k$ in a widely used masked average pooling [50] manner:

$$p_{o_k} = \frac{1}{|\Omega_{o_k}|} \sum_{x_j \in \Omega_{o_k}} F_j \tag{7}$$

where $F_j \in \mathbb{R}^C$ represents the pixel-wise representation of pixel $x_j$. Accordingly, the prototype set $P = \{p_{o_1}, p_{o_2}, ..., p_{o_N}\}$ for all anomalies can be easily obtained. The rationale for constructing prototypes for each OOD instance, rather than merely enforcing the separation between ID category prototypes and all pixel-wise representations, is further elaborated in Appendix A.4.

As our goal is to separate OOD prototypes from ID representations, one intuitive approach is to ensure that each OOD prototype is far from each prototype of the ID category [2]. The prototype for the $c$-th ID category can be established based on the segmentation model by:

$$q_c = \frac{\sum_{x_j} F_j \cdot \mathbb{1}[f_c(x_j) > \tau]}{\sum_{x_j} \mathbb{1}[f_c(x_j) > \tau]} \tag{8}$$

where $f_c(x_j)$ represents the probability of predicting pixel $x_j$ as category $c$, $\tau$ is a threshold for feature filtering, and $\mathbb{1}[\cdot]$ is an indicator function. Inspired by [13], we set a higher $\tau = 0.7$ to ensure the quality of each ID prototype. Accordingly, the prototype set for all ID categories in $x$ can be obtained as $Q = \{q_1, q_2, ..., q_K\}$.

Therefore, we formulate the ID semantic loss as:

$$L_{sem} = \frac{1}{N \cdot K} \sum_{p_{o_k} \in P} \sum_{q_c \in Q} cosSim[p_{o_k}, q_c] \tag{9}$$

where $cosSim[\cdot, \cdot]$ is the normalized cosine similarity (ranging $[0,1]$) between the prototype of each OOD instance and that of each ID category. In this way, the prototypes of OOD instances are separated from those of ID categories.

However, this strategy raises another issue: an OOD prototype may be close to one specific ID prototype while being far from others. We present Theorem 2 to provide a foundation for understanding this phenomenon.

**Theorem 2.** *(Finite-Category Mean Similarity Bound). Let $Q = \{q_c\}_{c=1}^K$ be a set of K ID class prototypes, and $p_{i_t}$ be an ID instance prototype from class t. Assume:*

*1. **Intra-class Alignment**: $cosSim(p_{i_t}, q_t) = 1 - \epsilon$, where $\epsilon \in [0,1]$ is small (e.g., $\epsilon \to 0^+$).*

*2. **Inter-class Separability**: $cosSim(p_{i_t}, q_c) \leq \delta$ for all $t \neq c$, where $\delta \in [0,1]$ is close to 0 (e.g., $\delta \to 0^+$).*

*Then, the mean cosine similarity $\bar{S}_K$ between $p_{i_t}$ and all ID prototypes is bounded by:*

$$\bar{S}_K := \frac{1}{K} \sum_{c=1}^K cosSim(p_{i_t}, q_c) \leq \frac{1 - \epsilon + (K-1)\delta}{K} \tag{10}$$

*For large K (e.g., $K \gg 1$), this simplifies to:*

$$\bar{S}_K \approx \delta + \frac{1 - \epsilon - \delta}{K} \xrightarrow{K \gg 1} \delta \tag{11}$$

The proof of Theorem 2 is provided in Appendix A. Theorem 2 indicates that using $L_{sem}$ cannot determine whether the prototype of an OOD instance falls into a specific ID category, especially when there are numerous ID categories (large $K$). To address this, we advocate for separating OOD prototypes from their nearest ID prototypes. Hence we calculate a nearest neighbor loss $L_{near}$ as:

$$L_{near} = \frac{1}{N \cdot M} \sum_{p_{o_k} \in P} \sum_{q_c \in Q_{o_k}} cosSim[p_{o_k}, q_c] \tag{12}$$

where $Q_{o_k}$ represents the top-$M$ nearest ID prototype neighbors of the OOD instance $o_k$.

Combining $L_{sem}$ and $L_{near}$, we formulate the overall instance-level feature separation loss as:

$$L_{feat} = \beta L_{sem} + (1 - \beta) L_{near} \tag{13}$$

where $\beta$ is a balanced factor to adjust $L_{sem}$ and $L_{near}$.

### 3.4 OOD Fine-tuning and Inference

**OOD fine-tuning**: Given a segmentation model trained on the close-set taxonomy, we propose the LNOIB framework for OOD fine-tuning to encourage the model to segment all OOD instances, as shown in Figure 2. The overall objective in LNOIB is formulated by:

$$L_{LNOIB} = \gamma_1 L_{pred} + \gamma_2 L_{feat} \tag{14}$$

where $\gamma_1$ and $\gamma_2$ are balanced factors. $L_{LNOIB}$ can be technically adapted to global-level objectives in existing OOD fine-tuning strategies. Specifically, $L_{pred}$ acts as an instance-level extension of the global-level objectives, while $L_{feat}$ can be easily computed using the latent features of segmentation models. As to the weights $\gamma_1$ and $\gamma_2$, please refer to Appendix E for further analysis.

**Inference**: After OOD fine-tuning with $L_{LNOIB}$, we use the segmentation model to generate anomaly scores. Different OOD fine-tuning strategies may provide various methods to yield anomaly scores. To ensure the versatility of LNOIB, we retain the original method used in each strategy. For example, if LNOIB is based on PEBAL [47], the anomaly score is obtained by calculating the free energy map as described in [47], without any modification. Consequently, the inference process depends on the specific strategy used and does not introduce extra computations, demonstrating the versatility and efficiency of LNOIB.

Table 1: Component-level results on multiple datasets. By incorporating LNOIB, existing OOD fine-tuning methods achieve higher results. The best results are in **bold** and the second best results are underlined. ↑ means higher values are better.

| Approach | SMIYC-RA | | | SMIYC-RO | | | FS Static | | | FS L&F | | | Road Anomaly | | |
|---|---|---|---|---|---|---|---|---|---|---|---|---|---|---|---|
| | sIoU ↑ | PPV ↑ | F1* ↑ | sIoU ↑ | PPV ↑ | F1* ↑ | sIoU ↑ | PPV ↑ | F1* ↑ | sIoU ↑ | PPV ↑ | F1* ↑ | sIoU ↑ | PPV ↑ | F1* ↑ |
| SynBoost (CVPR'21) [10] | 28.93 | 19.01 | 9.37 | 38.86 | 36.52 | 35.77 | 19.84 | 27.98 | 25.67 | 22.35 | 14.46 | 10.85 | 33.19 | 27.57 | 29.33 |
| FlowEneDet (UAI'23) [18] | 21.16 | 17.08 | 5.72 | 40.96 | 42.07 | 37.20 | 15.43 | 8.87 | 11.82 | 10.60 | 9.58 | 5.49 | 21.50 | 23.04 | 20.43 |
| RbA (ICCV'23) [36] | 56.26 | 41.35 | 42.04 | 47.44 | 56.16 | 50.42 | 37.03 | 30.96 | 26.94 | 27.47 | 18.69 | 20.27 | 37.19 | 35.70 | 42.27 |
| RPL (ICCV'23) [31] | 49.77 | 29.96 | 30.16 | 52.62 | 56.65 | 56.69 | 18.70 | 20.53 | 13.16 | 14.72 | 11.67 | 3.91 | 26.82 | 29.71 | 24.64 |
| CSL (AAAI'24) [58] | 45.14 | 47.70 | 44.85 | 41.66 | 48.98 | 46.27 | 27.81 | 23.55 | 18.90 | 22.03 | 12.64 | 6.79 | 25.18 | 33.67 | 27.64 |
| RWPM (ECCV'24) [57] | 53.10 | **58.25** | 47.44 | 52.89 | 70.21 | 64.85 | 34.52 | 37.15 | 28.21 | 19.86 | 26.27 | 18.31 | 41.65 | 44.14 | 45.38 |
| PixOOD (ECCV'24) [49] | 44.15 | 24.32 | 19.82 | 42.68 | 57.49 | 50.82 | 28.66 | 28.05 | 24.71 | 25.28 | 22.54 | 20.31 | 32.74 | 38.97 | 41.18 |
| EM (ICCV'21) [4] | 48.50 | 40.13 | 29.28 | 45.79 | 55.31 | 45.47 | 32.71 | 33.49 | 20.39 | 21.85 | 21.68 | 14.74 | 26.11 | 20.63 | 19.74 |
| +**LNOIB**(ours) | 58.73 | 47.42 | 43.90 | 50.86 | 58.80 | 48.57 | 44.38 | 37.20 | 37.79 | 34.74 | 29.56 | 23.97 | 35.72 | 33.18 | 27.30 |
| | (±0.26) | (±0.21) | (±0.35) | (±0.38) | (±0.22) | (±0.09) | (±0.30) | (±0.41) | (±0.25) | (±0.17) | (±0.24) | (±0.10) | (±0.18) | (±0.05) | (±0.29) |
| PEBAL (ECCV'22) [47] | 40.42 | 31.07 | 18.60 | 27.81 | 9.16 | 7.73 | 24.76 | 22.30 | 17.83 | 12.66 | 14.95 | 8.51 | 31.34 | 26.44 | 23.87 |
| +**LNOIB**(ours) | 52.17 | 39.55 | 33.48 | 32.88 | 12.90 | 13.46 | 38.84 | 29.75 | 28.44 | 27.36 | 35.78 | 17.73 | 37.68 | 30.91 | 32.65 |
| | (±0.41) | (±0.13) | (±0.44) | (±0.30) | (±0.06) | (±0.16) | (±0.25) | (±0.19) | (±0.09) | (±0.26) | (±0.33) | (±0.14) | (±0.33) | (±0.15) | (±0.12) |
| M2A (PAMI'24) [42] | 51.47 | 46.70 | 45.26 | 50.49 | 69.38 | 64.02 | 35.24 | 27.70 | 29.18 | 26.51 | 19.43 | 23.76 | 47.43 | 40.80 | 44.57 |
| +**LNOIB**(ours) | **63.15** | 57.37 | **60.08** | **61.50** | **73.71** | **70.25** | **52.08** | **44.93** | **42.96** | **39.16** | **38.10** | **34.80** | **53.70** | **51.46** | **50.97** |
| | (±0.48) | (±0.53) | (±0.27) | (±0.07) | (±0.51) | (±0.46) | (±0.18) | (±0.20) | (±0.30) | (±0.31) | (±0.17) | (±0.19) | (±0.38) | (±0.25) | (±0.08) |

## 4 Experiments

### 4.1 Experimental Setup

**Datasets**: As to the ID dataset, we adopt the Cityscapes dataset [9] for pre-training, which includes 2975 training and 500 validation images, containing 19 different urban scene categories. For OOD datasets, we evaluate our approach on various AS benchmarks. The Fishyscapes benchmark [1] includes two datasets: Fishyscapes Static (FS Static) and Fishyscapes Lost & Found (FS L&F). The former contains 30 validation images from blending Pascal [12], and the latter is based on Lost and Found dataset [39], with 100 validation images. SMIYC benchmark [3] consists of two separate datasets: RoadAnomaly (SMIYC-RA) and RoadObstacle (SMIYC-RO), which contain 10 and 30 validation images with road anomalies and obstacles, respectively. Additionally, the Road Anomaly dataset [30], which served as a precursor to SMIYC, includes 60 images with anomalies located in or near the road for validation.

**Evaluation Metrics**: As our target is to segment all anomalies regardless of their sizes, we mainly focus on the component-level metrics to evaluate LNOIB, including the component-wise intersection over union (sIoU), the positive predictive value (PPV), and the averaged component-wise F1 score (F1*). Please refer to Appendix C for detailed information. These metrics reflect the extent to which each disjoint component is covered, ensuring that smaller components are weighted equally with larger ones. Furthermore, we also consider the commonly used pixel-level metrics, i.e., Area under the Precision-Recall Curve (AuPRC) and False Positive Rate at a true positive rate of 95% ($FPR_{95}$), to guarantee the overall pixel-level quality. In addition, inspired by [37], we also adopt instance-level metrics iAP and iAP50 to further validate the effectiveness of our method. For detailed definitions and corresponding results, please refer to Appendix F.

**Implementation Details**: To test the versatility of our approach, we employ LNOIB to EM [4], PEBAL [47], and M2A [41], for the three leading AS solutions utilizing global-level OOD fine-tuning strategies. For fair comparisons, we adopt the same configurations as each respective approach during the first stage, where a closed-set segmentation model is trained on the Cityscapes [9]. During the OOD fine-tuning stage, our LNOIB objectives are built on the global-level losses used in EM, PEBAL, and M2A, respectively, with parameters set as $\alpha = 0.5$, $\beta = 0.5$, $M = 1$, $\tau = 0.7$, and $\gamma_1 = \gamma_2 = 1$, empirically. For detailed formulations of LNOIB objective, please refer to Appendix B. We select features from stages 2, 3, and 4 of each backbone, upsample them to 1/4 of the image size, and incorporate them to calculate $L_{feat}$. For consistency, we fine-tune the whole segmentation model for each method using their corresponding configurations. We adopt AnomalyMix [47] to sample 297 images from COCO [28] and mix them into Cityscapes to generate outlier images and identify each OOD instance for calculating $L_{ins}$ and $L_{feat}$. During inference, we follow the methods to yield anomaly scores in EM, PEBAL, and M2A, respectively.

### 4.2 Main Results

**Component-level performance**: We first evaluate the component-level performance of LNOIB on multiple datasets. The results for sIoU, PPV, and F1* are shown in Table 1. As seen, OOD fine-tuning strategies like EM, PEBAL, and M2A show significant improvements when integrated with LNOIB. Notably, M2A, when extended with LNOIB, outperforms other methods such as SynBoost [10], FlowEneDet [18], CSL [58], RbA [36], RPL [31], RWPM [57], and PixOOD [49]. Please note that

Table 2: Pixel-level performances on SMIYC-RA, SMIYC-RO, FS Static, FS L&F, and Road Anomaly. By incorporating LNOIB, existing OOD fine-tuning strategies achieve improvements in most cases. ↓ means lower values are better.

| Approach | SMIYC-RA | | SMIYC-RO | | FS Static | | FS L&F | | Road Anomaly | |
|---|---|---|---|---|---|---|---|---|---|---|
| | AuPRC↑ | FPR$_{95}$↓ | AuPRC↑ | FPR$_{95}$↓ | AuPRC↑ | FPR$_{95}$↓ | AuPRC↑ | FPR$_{95}$↓ | AuPRC↑ | FPR$_{95}$↓ |
| SynBoost (CVPR'21) [10] | 50.64 | 57.63 | 58.89 | 8.47 | 48.44 | 47.71 | 40.99 | 34.47 | 41.83 | 59.72 |
| FlowEneDet (UAI'23) [18] | 52.61 | 61.43 | 76.04 | 1.38 | 52.61 | 14.91 | 56.11 | 3.87 | 76.35 | 15.24 |
| RbA (ICCV'23) [36] | 86.13 | 15.94 | 87.85 | 3.33 | 83.26 | **4.22** | 60.96 | 10.63 | 78.45 | **11.83** |
| RPL (ICCV'23) [31] | 72.60 | 12.65 | 75.53 | 4.19 | 87.27 | 5.69 | 49.92 | 16.78 | 63.96 | 26.18 |
| CSL (AAAI'24) [58] | 76.75 | **9.14** | 81.55 | 0.96 | 79.73 | 6.34 | 70.68 | 8.15 | 61.38 | 43.80 |
| RWPM (ECCV'24) [57] | 88.30 | 11.87 | 91.68 | 0.40 | 81.69 | 4.51 | 74.45 | 4.83 | 76.12 | 13.67 |
| PixOOD (ECCV'24) [49] | 58.83 | 34.64 | 84.96 | 0.93 | 85.37 | 4.77 | 84.53 | 2.27 | 75.79 | 13.38 |
| EM (ICCV'21) [4] | 83.95 | 17.13 | 84.81 | 2.36 | 86.56 | 9.37 | 80.22 | 5.40 | 70.86 | 21.47 |
| +**LNOIB(ours)** | 86.26 (±0.47) | 15.28 (±0.11) | 84.47 (±0.35) | 2.39 (±0.06) | **86.95** (±0.21) | 8.89 (±0.10) | 81.68 (±0.24) | 5.18 (±0.07) | 72.42 (±0.68) | 19.74 (±0.17) |
| PEBAL (ECCV'22) [47] | 53.72 | 32.18 | 27.24 | 19.58 | 82.73 | 6.81 | 59.83 | 6.49 | 62.37 | 28.29 |
| +**LNOIB(ours)** | 61.46 (±0.44) | 22.03 (±0.18) | 31.30 (±0.07) | 18.44 (±0.14) | 83.81 (±0.52) | 4.47 (±0.03) | 64.50 (±0.40) | 3.82 (±0.11) | 65.35 (±0.14) | 22.77 (±0.10) |
| M2A (PAMI'24) [42] | 87.83 | 15.09 | 91.52 | 0.43 | 71.36 | 10.28 | 89.91 | 1.85 | 75.70 | 16.31 |
| +**LNOIB(ours)** | **92.20** (±0.11) | 11.34 (±0.07) | **92.58** (±0.42) | **0.33** (±0.04) | 75.19 (±0.55) | 8.27 (±0.11) | **90.54** (±0.58) | **1.68** (±0.05) | **79.38** (±0.22) | 13.29 (±0.08) |

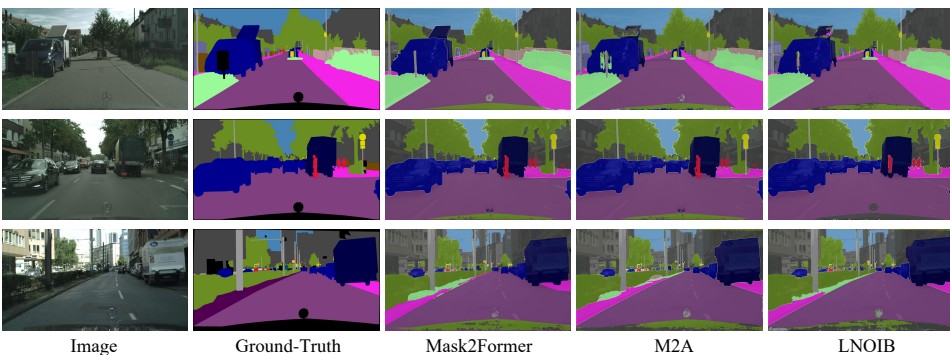

| Image | Ground-Truth | Mask2Former | M2A | LNOIB |

Figure 3: Qualitative results on Cityscapes. We compare the close-set performance of Mask2Former, Mask2Former fine-tuned with M2A, and Mask2Former fine-tuned with LNOIB (based on M2A), showing minimal close-set performance drop.

LNOIB is compatible with existing OOD fine-tuning approaches. Therefore, we select EM, PEBAL, and M2A as the baseline methods for integration. For a more precise assessment of the performance gains brought by our instance-level extension, we report our own reproduced results of these baselines to exclude the influence of setup discrepancies (such as hardware discrepancy and random seed discrepancy, as we report the mean of 3 runs). These results highlight that LNOIB improves AS by comprehensively capturing more anomalies. To further isolate the impact on large anomalies, we need to analyze pixel-level metrics. If pixel-level improvement is less pronounced, it would indicate that LNOIB primarily enhances small anomaly segmentation.

**Instance-level performance**: Integrating LNOIB into existing OOD fine-tuning strategies significantly improves instance-level performance. For more details, please refer to Appendix F, which further demonstrates that LNOIB enhances segmentation across all anomalies.

**Pixel-level performance**: LNOIB is primarily designed to enhance component-level and instance-level results but also delivers promising results in pixel-level metrics. Specifically, we evaluate AuPRC and FPR$_{95}$ metrics across multiple datasets, as summarized in Table 2. The results show that incorporating LNOIB improves AuPRC and FPR$_{95}$ for most OOD fine-tuning approaches, including EM, PEBAL, and M2A. While these improvements are more pronounced at the component and instance levels, pixel-level gains appear relatively modest. This subtle improvement suggests that the enhancements are primarily driven by better handling of small anomalies. Furthermore, the competitive pixel-level results indicate that OOD fine-tuning with LNOIB effectively preserves the high quality of large OOD instances. This is because large anomalies contribute substantially to pixel-level scores, any degradation in their segmentation would have led to a noticeable drop in overall performance.

**Close-set segmentation**: OOD fine-tuning may degrade the performance on ID categories. To investigate this, we evaluate the closed-set performance of segmentation models on Cityscapes after

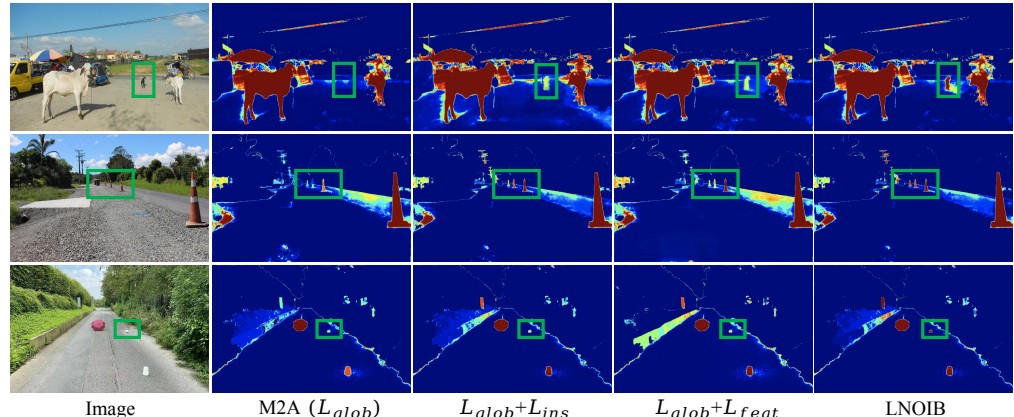

| Image | M2A ($L_{glob}$) | $L_{glob}+L_{ins}$ | $L_{glob}+L_{feat}$ | LNOIB |
|---|---|---|---|---|

Figure 4: Qualitative results of applying LNOIB to M2A. Instances that do not belong to the Cityscapes categories are regarded as anomalies. Vanilla $L_{glob}$ in M2A falls short in predicting small anomalies (see green boxes). With the combination of $L_{ins}$ and $L_{feat}$ in LNOIB, it yields competitive performances on small OOD instances.

Table 3: Ablation results of each component in prediction-based objective and feature separation objective on SMIYC-RA. Incorporating $L_{ins}$, $L_{sem}$, and $L_{near}$ contributes to further improvements.

(a) Prediction-based Objective

| Approach | $L_{glob}$ | $L_{ins}$ | sIoU ↑ | PPV ↑ | F1* ↑ | AuPRC ↑ | FPR$_{95}$ ↓ |
|---|---|---|---|---|---|---|---|
| EM | ✓ | | 48.50 | 40.13 | 29.28 | 83.95 | 17.13 |
| | | ✓ | 44.29 | 37.06 | 26.84 | 53.60 | 35.15 |
| | ✓ | ✓ | **55.79** | **46.36** | **39.20** | **85.10** | **17.02** |
| PEBAL | ✓ | | 40.42 | 31.07 | 18.60 | 53.72 | 32.18 |
| | | ✓ | 31.93 | 26.69 | 14.37 | 41.16 | 44.70 |
| | ✓ | ✓ | **50.60** | **36.95** | **27.44** | **58.98** | **27.59** |
| M2A | ✓ | | 51.47 | 46.70 | 45.26 | 87.83 | 15.09 |
| | | ✓ | 47.72 | 44.16 | 42.09 | 66.32 | 31.90 |
| | ✓ | ✓ | **62.82** | **53.55** | **54.86** | **88.07** | 15.47 |

(b) Feature Separation Objective

| Approach | $L_{sem}$ | $L_{near}$ | sIoU ↑ | PPV ↑ | F1* ↑ | AuPRC ↑ | FPR$_{95}$ ↓ |
|---|---|---|---|---|---|---|---|
| EM | | | 48.50 | 40.13 | 29.28 | 83.95 | 17.13 |
| | ✓ | | 42.18 | 35.62 | 27.21 | 75.74 | 18.87 |
| | | ✓ | 48.71 | 41.10 | 29.16 | 81.57 | 16.88 |
| | ✓ | ✓ | **53.94** | **43.61** | **38.77** | **85.53** | **16.70** |
| PEBAL | | | 40.42 | 31.07 | 18.60 | 53.72 | 32.18 |
| | ✓ | | 33.03 | 25.85 | 14.47 | 46.60 | 37.57 |
| | | ✓ | 40.78 | 30.56 | 18.83 | 55.01 | 30.78 |
| | ✓ | ✓ | **44.35** | **34.76** | **22.13** | **56.41** | **26.89** |
| M2A | | | 51.47 | 46.70 | 45.26 | 87.83 | 15.09 |
| | ✓ | | 44.85 | 42.73 | 40.51 | 82.41 | 19.83 |
| | | ✓ | 53.03 | 48.43 | 44.97 | 84.79 | 15.17 |
| | ✓ | ✓ | **57.75** | **50.67** | **48.79** | **85.70** | **13.28** |

applying OOD fine-tuning with LNOIB. Interestingly, LNOIB maintains competitive mIoU scores (within a 1% drop compared to vanilla segmentation models, please see Appendix D for details). For a clearer understanding, qualitative results are also presented in Figure 3, showing that OOD fine-tuning with LNOIB preserves strong ID performance on Cityscapes, with results that remain comparable to those of the original Mask2Former.

## 4.3 Ablation Study

We investigate the impact of $L_{pred}$ and $L_{feat}$ in LNOIB. To achieve this, we conduct thorough ablation experiments to determine the optimal settings on EM, PEBAL, and M2A using the SMIYC-RA dataset, which includes OOD instances of various sizes, from large to small. Moreover, to evaluate the generality of these components, we also conduct extensive ablation study on multiple datasets. Please refer to Appendix E for more details.

**Global vs. Instance-Level Objective**: We explore the influence of the commonly used global-level objective $L_{glob}$ and our proposed instance-level objective $L_{ins}$ for OOD fine-tuning on overall performances. Specifically, we set $\alpha = 1$, $\alpha = 0$, and $\alpha = 0.5$ in Eq. (6), which correspond to solely adopting the global-level objective $L_{glob}$ (equivalent to the vanilla OOD fine-tuning strategy), merely employing the instance-level objective $L_{ins}$, and using a hybrid objective of $L_{glob}$ and $L_{ins}$, respectively. We remove the irrelevant feature separation objective here, and the results are presented in Table 3a. As shown in the table, combining $L_{glob}$ and $L_{ins}$ achieves the best performance across multiple metrics. We also observe that solely using $L_{ins}$ even results in decreases in both metrics, due to its insufficient attention to large anomalies compared to $L_{glob}$. Consequently, the overall prediction-based objective that incorporates both $L_{glob}$ and $L_{ins}$ effectively addresses both large and small anomalies, demonstrating promising performances.

Furthermore, we explore the optimal factor $\alpha$ in Eq. (6) with respect to the performance. The results and analyses are provided in Appendix E.

**Features Separation Objective**: We explore the impact of feature separation objective for OOD fine-tuning. We directly incorporate $L_{sem}$ and $L_{near}$ with the commonly used $L_{glob}$ in each respective OOD fine-tuning strategy, and the results are shown in Table 3b. The table shows that using $L_{sem}$ alone even decreases the overall perfomance. Solely adopting $L_{sem}$ may cause the OOD prototype to align with the prototype of a specific ID category, as presented in Theorem 2. On the other hand, solely using $L_{near}$ is also insufficient because it cannot ensure that the OOD prototype stays separated from all ID prototypes, especially when several ID categories share similar representations. Therefore, $L_{sem}$ and $L_{near}$ are complementary, and their combination achieves the best results.

Moreover, we aim to identify the optimal balance factor $\beta$. The results and analyses are provided in Appendix E. Additionally, we investigate the impact of the threshold $\tau$ and the number of nearest neighbors $M$ for $L_{feat}$. For more details, please also refer to Appendix E.

### 4.4 Qualitative Results

We further explore the qualitative results of LNOIB to show the effectiveness of each proposed objective. Figure 4 illustrates how each objective refines the performance of M2A. Specifically, we compare the vanilla M2A (solely using $L_{glob}$) with M2A+$L_{ins}$ ($L_{pred}$), M2A+$L_{feat}$, and M2A with LNOIB (using the OOD fine-tuning objective in Eq. (14)) for OOD fine-tuning. As depicted in the figure, the vanilla M2A falls short in capturing small anomalies. With the addition of $L_{ins}$ and $L_{feat}$, LNOIB achieves superior segmentation performances on several small OOD instances (e.g., small cow, traffic cones, and cup in the first, second, and third rows, respectively), while maintaining competitive results on larger anomalies. When using the overall objective in LNOIB, M2A achieves the best performance.

## 5 Conclusion

In this paper, we proposed a novel instance-level OOD fine-tuning framework for AS, dubbed LNOIB, which is designed to effectively segment OOD instances regardless of their sizes. We provided a theoretical analysis explaining why current OOD fine-tuning strategies struggle to detect small anomalies. Building on this insight, we introduced a simple yet effective instance-level objective to target small anomalies. Furthermore, we proposed a feature separation objective to further enhance the segmentation of small anomalies. Note that, beyond a single objective design, LNOIB serves as a versatile paradigm that can be seamlessly integrated with existing OOD fine-tuning strategies, without introducing extra cost during inference. Extensive experimental results show that incorporating LNOIB into existing OOD fine-tuning strategies yields superior performance.

## 6 Limitation and Future Work

LNOIB serves as a versatile mechanism for existing OOD fine-tuning strategies, demonstrating strong performance across EM, PEBAL, and M2A with a unified set of hyperparameters. However, for other approaches, particularly those whose global-level objective values are significantly larger or smaller than the feature separation term, hyperparameter tuning may be necessary to achieve optimal results. Although LNOIB effectively refines small anomalies in most cases, the quality is not always perfect, as illustrated in the first row of Figure 4. Therefore, there remains a potential risk when deploying it in safety-critical applications. Moreover, since LNOIB serves as an extension of existing OOD fine-tuning approaches, our current evaluation is conducted on validation images. Further experiments on more challenging test sets are expected to provide a more comprehensive assessment of its generalization capability. Finally, OOD fine-tuning itself remains controversial, as it only mimics limited OOD scenarios and fails to capture the full diversity of real-world OOD patterns. Therefore, we believe it is worthwhile to explore alternative approaches for addressing this issue.

For future work, we plan to extend our instance-level framework to other segmentation scenarios to investigate whether this mechanism can improve the segmentation of small components across different tasks [61, 45, 7, 59, 53]. Specifically, although there are several hyperparameters in LNOIB, they also provide different selections to suit the corresponding tasks. Moreover, extending this framework to video tasks [23, 11, 14, 15, 51] may also be a promising direction for future research.

## Acknowledgments

This work was supported by the National Natural Science Foundation of China (No. 62172385), the Natural Science Foundation of Jiangsu Province (BK20241819), the Innovation Program for Quantum Science and Technology (No. 2021ZD0302900), CCF-Baidu Open Fund, Natural Science Foundation of Jiangsu Province (BK20240463), Fundamental Research Funds for the Central Universities, and the Anhui Provincial Department of Science and Technology under Grant 202103a05020009.

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

# Supplementary Material

## A    More Proofs

### A.1    Proof of Lemma 1

**Lemma 1.** *(Weighted Loss Decomposition). The total loss $L_{out}$ decomposes into instance-specific components:*

$$L_{out} = \frac{1}{|\Omega_{out}|} \sum_{o_k \in O} |\Omega_{o_k}| \cdot \underbrace{\mathbb{E}_{x_j \sim \Omega_{o_k}} [l_{out} [s(x_j), y_j]]}_{L_{o_k}} \triangleq \sum_{k=1}^{N} w_k L_{o_k} \tag{15}$$

*where $w_k \triangleq \frac{|\Omega_{o_k}|}{|\Omega_{out}|}$ is the normalized weight of instance $o_k$ and $L_{o_k}$ is the mean loss over $\Omega_{o_k}$.*

*Proof.* As $\sum_{k=1}^{N} |\Omega_{o_k}| = |\Omega_{out}| < +\infty$ and $\Omega_{o_1} \cap \Omega_{o_2} \cap ... \cap \Omega_{o_N} = \emptyset$, the expectation over $\Omega_{out}$ splits into disjoint domains:

$$L_{out} = \frac{1}{|\Omega_{out}|} \sum_{x_j \in \Omega_{out}} l_{out}[s(x_j), y_j] \tag{16}$$

$$= \frac{1}{|\Omega_{out}|} \sum_{o_k \in O} \sum_{x_j \in o_k} l_{out}[s(x_j), y_j] \tag{17}$$

$$= \frac{1}{|\Omega_{out}|} \sum_{o_k \in O} |\Omega_{o_k}| \cdot \mathbb{E}_{x_j \sim \Omega_{o_k}} [l_{out} [s(x_j), y_j]] \tag{18}$$

$$\square$$

### A.2    Proof of Theorem 1

**Theorem 1.** *(Dominant Instance Effect). If there exists $t \in \{1, ..., N\}$, such that:*

$$|\Omega_t| \geq (1 - \epsilon)|\Omega_{out}| \quad \text{for some } \epsilon \in (0, 1), \tag{19}$$

*then the loss $L_{out}$ is bounded by:*

$$(1 - \epsilon)L_{o_t} \leq L_{out} \leq (1 - \epsilon)L_{o_t} + \epsilon \cdot \max_{k \neq t} L_{o_k} \tag{20}$$

*Proof.* From Lemma 1 and the weight condition, we have the lower bound:

$$L_{out} = w_t L_{o_t} + \sum_{k \neq t} w_k L_{o_k} \geq w_t L_{o_t} \geq (1 - \epsilon)L_{o_t}. \tag{21}$$

As to the upper bound:

$$L_{out} \leq w_t L_{o_t} + \left( \sum_{k \neq t} w_k \right) \max_{k \neq t} L_{o_k} \leq (1 - \epsilon)L_{o_t} + \epsilon \cdot \max_{k \neq t} L_{o_k} \tag{22}$$

$$\square$$

## A.3 Proof of Theorem 2

**Theorem 2.** *(Finite-Category Mean Similarity Bound). Let $Q = \{q_c\}_{c=1}^K$ be a set of $K$ ID class prototypes, and $p_{i_t}$ be an ID instance prototype from class $t$. Assume:*

**1. Intra-class Alignment**: *$cosSim(p_{i_t}, q_t) = 1 - \epsilon$, where $\epsilon \in [0, 1]$ is small (e.g., $\epsilon \to 0^+$).*

**2. Inter-class Separability**: *$cosSim(p_{i_t}, q_c) \leq \delta$ for all $t \neq c$, where $\delta \in [0, 1]$ is close to 0 (e.g., $\delta \to 0^+$).*

*Then, the mean cosine similarity $\bar{S}_K$ between $p_{i_t}$ and all ID prototypes is bounded by:*

$$\bar{S}_K := \frac{1}{K} \sum_{c=1}^K cosSim(p_{i_t}, q_c) \leq \frac{1 - \epsilon + (K - 1)\delta}{K} \tag{23}$$

*For large $K$ (e.g., $K \gg 1$), this simplifies to:*

$$\bar{S}_K \approx \delta + \frac{1 - \epsilon - \delta}{K} \xrightarrow{K \gg 1} \delta \tag{24}$$

*Proof.* The sum of $\bar{S}_K$ can be decomposed as:

$$
\begin{aligned}
\bar{S}_K &= \frac{1}{K} \left( cosSim(p_{i_t}, q_t) + \sum_{c \neq t} cosSim(p_{i_t}, q_c) \right) \\
&= \frac{1}{K} \left( 1 - \epsilon + \sum_{c \neq t} cosSim(p_{i_t}, q_c) \right) \\
&\leq \frac{1}{K} \left( 1 - \epsilon + (K - 1)\delta \right)
\end{aligned}
\tag{25}
$$

$\square$

**More explanations**: In a well-trained segmentation model, the intra-class feature variation tends to be minimal. Consequently, the prototype of an ID instance is typically close to the center of its corresponding category. However, this raises a concern: relying solely on ID semantic loss may cause an OOD instance to be mistakenly classified as belonging to one of the ID categories, especially when the number of ID categories is large. This occurs because the semantic loss for ID categories can also be near zero under such circumstances, which gives a false sense that the prototype of an OOD instance is far from all the ID semantic prototypes. Thus we also need the nearest neighbor loss to deal with this issue, as demonstrated in the main paper.

## A.4 Instance-Level OOD Prototypes

In the main paper, we propose to construct an OOD prototype for each OOD instance, rather than merely enforcing the separation between ID category prototypes and all pixel-wise OOD representations. This design choice is motivated by the limitations of global-level feature separation objective, which tends to overlook small or localized anomalies.

We provide a theoretical analysis of the feature separation objective $L_{sem}$ when globally separating each OOD pixel-wise representation from ID semantic prototypes (Averaging all OOD pixels suffers from the same issue). This analysis shows that, without instance-level OOD prototypes, small anomalies are likely to be ignored in the feature separation objective.

Firstly, we formulate the global-level ID semantic objective $L_{gsem}$ as:

$$L_{gsem} = \frac{1}{K|\Omega_{out}|} \sum_{q_c \in Q} \sum_{x_j \in \Omega_{out}} cosSim[F_{x_j}, q_c] \tag{26}$$

The objective of $L_{gsem}$ is to enforce clear separation between ID category prototypes and pixel-wise OOD representations.

**Lemma 2.** *The total objective of $L_{gsem}$ decomposes into instance-specific components:*

$$L_{gsem} = \frac{1}{K|\Omega_{out}|} \sum_{q_c \in Q} \sum_{x_j \in \Omega_{out}} cosSim[F_{x_j}, q_c] \tag{27}$$

$$= \frac{|\Omega_{o_k}|}{|\Omega_{out}|} \sum_{o_k \in O} \underbrace{\frac{1}{K} \mathbb{E}_{x_j \sim \Omega_{o_k}} \sum_{q_c \in Q} cosSim[F_{x_j}, q_c]}_{L_{gsem,o_k}} \tag{28}$$

$$\triangleq \sum_{k=1}^{N} r_k \cdot L_{gsem,o_k} \tag{29}$$

*where $r_k = \frac{|\Omega_k|}{|\Omega_{out}|}$ is the normalized weight of instance, and $L_{gsem,o_k}$ is the mean ID semantic loss over $\Omega_{o_k}$.*

*Proof.*

$$L_{gsem} = \frac{1}{K|\Omega_{out}|} \sum_{q_c \in Q} \sum_{x_j \in \Omega_{out}} cosSim[F_{x_j}, q_c] \tag{30}$$

$$= \frac{1}{K|\Omega_{out}|} \sum_{q_c \in Q} \sum_{o_k \in O} \sum_{x_j \in o_k} cosSim[F_{x_j}, q_c] \tag{31}$$

$$= \frac{1}{K|\Omega_{out}|} \sum_{q_c \in Q} \sum_{o_k \in O} |\Omega_{o_k}| \cdot \mathbb{E}_{x_j \sim \Omega_{o_k}} cosSim[F_{x_j}, q_c] \tag{32}$$

Based on the definition of $L_{gsem,o_k}$, we have:

$$L_{gsem} = \frac{1}{|\Omega_{out}|} \sum_{o_k \in O} |\Omega_{o_k}| \cdot L_{gsem,o_k} \tag{33}$$

$\square$

**Theorem 3.** *If there exists $t \in \{1, 2, ..., N\}$, such that:*

$$|\Omega_t| \geq (1 - \epsilon)|\Omega_{out}| \quad \text{for some } \epsilon \in (0, 1), \tag{34}$$

*then $L_{gsem}$ is bounded by:*

$$(1 - \epsilon)L_{gsem,o_t} \leq L_{gsem} \leq (1 - \epsilon)L_{gsem,o_t} + \epsilon \cdot \max_{k \neq t} L_{gsem,o_k}. \tag{35}$$

*Proof.* The proof follows a reasoning analogous to that of Theorem 1. $\square$

**Corollary 2.** *If $\frac{|\Omega_t|}{|\Omega_{out}|} \to 1$, then $L_{gsem} \to L_{gsem,o_t}$ in probability.*

*Proof.* This proof mirrors the approach taken in Corollary 1. $\square$

Based on the theoretical analysis above, globally separating each pixel-wise OOD feature from ID prototypes also overlooks the optimization for small anomalies, thus our ID semantic objective $L_{sem}$ in Eq. (9) is more reasonable, which equally optimizes each anomaly. Moreover, the theoretical analysis of using $L_{near}$ is the same. Therefore, the feature separation objective also benefits from our instance-level framework to refine the optimization on small anomalies.

# B How LNOIB Adapts to Existing OOD Fine-Tuning Strategies

To provide a clearer understanding of how to apply LNOIB to current OOD fine-tuning strategies that employ global-level objectives, we present a detailed formulation of the integration of EM [4] with LNOIB, PEBAL [47] with LNOIB, and M2A [41] with LNOIB, respectively.

## B.1 Global vs. Instance-Level Objective

As shown in the main paper, the commonly used global-level objective $L_{glob}$ and our proposed instance-level objective $L_{ins}$ can be formulated as follows:

$$L_{glob} = \underbrace{\frac{1}{N_{in}} \sum_{x_i \in \Omega_{in}} l_{in}[s(x_i), y_i]}_{L_{in}} + \underbrace{\frac{1}{N_{out}} \sum_{x_j \in \Omega_{out}} l_{out}[s(x_j), y_j]}_{L_{out}} \tag{36}$$

$$L_{ins} = L_{in} + \frac{1}{N} \sum_{o_k \in O} \sum_{x_j \in \Omega_{o_k}} \frac{1}{|\Omega_{o_k}|} \cdot l_{out}[s(x_j), y_j] \tag{37}$$

where $\Omega_{in}$ and $\Omega_{out}$ represent the image lattices of ID and OOD regions (based on the ground-truth binary mask $y$), respectively. $N_{in} = |\Omega_{in}|$ and $N_{out} = |\Omega_{out}|$ separately are the total number of pixels in $\Omega_{in}$ and $\Omega_{out}$. $O$ denotes the set of OOD instances and $\Omega_{o_k}$ is the image lattice of the OOD instance $o_k$. These notations are also introduced in the main paper.

The proposed instance-level objective $L_{ins}$ does not refer to a specific loss function. Instead, it acts as a versatile function that can be extended to the existing global-level objective $L_{glob}$ for OOD fine-tuning. Specifically, the concrete formulation of $L_{ins}$ is determined by $l_{in}$ and $l_{out}$ used in the corresponding $L_{glob}$. Note that, the formulation of $l_{in}$ and $l_{out}$ in Eq. (1) is for high-level understanding by comparing the similarity between the predicted anomaly score and the corresponding ground-truth. For each specific implementation, the input parameters of $l_{in}$ and $l_{out}$ may vary. For example, EM directly adopts the predicted softmax distribution $f(x)$, which acts as an indicator of anomaly score, to calculate the loss. Consequently, we omit the input parameter lists of $l_{in}$ and $l_{out}$ below for clarity.

### B.1.1 LNOIB on Entropy Maximization

Given a mixed-content image $x \in \mathbb{R}^{3 \times H \times W}$ (containing both ID and OOD regions), we first employ the well-trained semantic segmentation model DeepLabv3+ [5] to generate a softmax prediction $f(x) \in \mathbb{R}^{K \times H \times W}$ for close-set categories, where $K$ is the number of ID categories. Then, according to the close-set mask $y^{close} \in \{1, 2, ..., K\}^{H \times W}$, $l_{in}^{EM}$ and $l_{out}^{EM}$ are formulated as:

$$\begin{cases} l_{in}^{EM} = -\sum_{c=1}^{K} \log(f_c(x_i)) \cdot \mathbb{1}[c = y_i^{close}] \\ l_{out}^{EM} = -\frac{1}{K} \sum_{c=1}^{K} \log(f_c(x_j)) \end{cases} \tag{38}$$

where $\mathbb{1}[\cdot]$ is an indicator function. Note that $l_{in}$ is a typical cross-entropy loss used in close-set semantic segmentation, while $l_{out}^{EM}$ is an upper bound of the softmax entropy according to Jensen's inequality (for more details, please refer to [4]). The goal for minimizing $l_{in}^{EM}$ is to make each pixel predict a higher score for the corresponding ID category, while the target for minimizing $l_{out}^{EM}$ is to let the predicted results follow a uniform distribution. Both of these losses measure the similarity between the indicator of anomaly scores and the corresponding binary ground-truth mask, resulting in a well-trained boundary of ID and OOD regions after OOD fine-tuning.

Accordingly, the vanilla global-level objective $L_{glob}^{EM}$ is calculated by:

$$L_{glob}^{EM} = \frac{1}{N_{in}} \sum_{x_i \in \Omega_{in}} l_{in}^{EM} + \frac{1}{N_{out}} \sum_{x_j \in \Omega_{out}} l_{out}^{EM} \qquad (39)$$

To adapt the instance-level objective to $L_{glob}^{EM}$, we first divide $\Omega_{out}$ into several separate OOD instances: $\Omega_{out} = \Omega_{o_1} \cup \Omega_{o_2} \cup ... \cup \Omega_{o_N}$, as demonstrated in the main paper. Building on the global-level objective $L_{glob}^{EM}$, we then formulate the instance-level objective $L_{ins}^{EM}$ for EM as:

$$L_{ins}^{EM} = \frac{1}{N_{in}} \sum_{x_i \in \Omega_{in}} l_{in}^{EM} + \frac{1}{N} \sum_{o_k \in O} \sum_{x_j \in \Omega_{o_k}} \frac{1}{|\Omega_{o_k}|} \cdot l_{out}^{EM} \qquad (40)$$

In this manner, we adapt the instance-level objective to EM, followed by the calculation of $L_{pred}^{EM}$:

$$L_{pred}^{EM} = \alpha L_{glob}^{EM} + (1 - \alpha) L_{ins}^{EM} \qquad (41)$$

where $\alpha$ is a balanced factor as demonstrated in the main paper.

### B.1.2 LNOIB on PEBAL

Unlike EM, PEBAL employs a $K + 1$ class semantic segmentation model, where the class $K + 1$ is designated for the anomaly score. In the case of a mixed-content image, if a pixel $i$ belongs to an inlier class, its ground-truth is $y_i \in \{1, 2, ..., K\}$, which corresponds to a typical one-hot encoded vector. On the other hand, if a pixel $j$ is an outlier, we assign a uniform value 1 across all the closed-set categories (similar to EM). This is because we need to calculate the close-set predicted distribution that also reflects the anomaly score for optimization.

Therefore, $l_{in}^{PEBAL}$ and $l_{out}^{PEBAL}$ can be calculated as:

$$\begin{cases} l_{in}^{PEBAL} = -\log \left( \sum_{c=1}^{K} (f_c(x_i)) \cdot \mathbb{1}[c = y_i^{close}] + \frac{f_{K+1}(x_i)}{a_i} \right) \\ l_{out}^{PEBAL} = -\log \left( \frac{1}{K} \sum_{c=1}^{K} (f_c(x_i)) + \frac{f_{K+1}(x_j)}{a_j} \right) \end{cases} \qquad (42)$$

where $a$ is an energy-biased pixel-wise adaptive factor to penalize an outlier pixel if it is predicted as a close-set category. On the other hand, $a$ encourages inlier pixels to make a close-set prediction. This loss is known as Gambler loss, and $a$ is obtained by calculating the pixel-wise free energy. Please refer to [47] and [32] for more details about abstention learning. Minimizing $l_{in}^{PEBAL}$ and $l_{out}^{PEBAL}$ also aims to bring the prediction of the anomaly score closer to the corresponding binary ground truth. Note that, there are also two auxiliary terms in $l_{in}^{PEBAL}$ and $l_{out}^{PEBAL}$. We omit the formulation of them here for convenience, as these two terms contribute little to the overall performance.

Accordingly, the global-level objective for PEBAL is calculated as:

$$L_{glob}^{PEBAL} = \frac{1}{N_{in}} \sum_{x_i \in \Omega_{in}} l_{in}^{PEBAL} + \frac{1}{N_{out}} \sum_{x_j \in \Omega_{out}} l_{out}^{PEBAL} \qquad (43)$$

Based on the formulation of $l_{in}^{PEBAL}$ and $l_{out}^{PEBAL}$, our instance-level objective for PEBAL is calculated by:

$$L_{ins}^{PEBAL} = \frac{1}{N_{in}} \sum_{x_i \in \Omega_{in}} l_{in}^{PEBAL} + \frac{1}{N} \sum_{o_k \in O} \sum_{x_j \in \Omega_{o_k}} \frac{1}{|\Omega_{o_k}|} \cdot l_{out}^{PEBAL} \qquad (44)$$

Combining $L_{glob}^{PEBAL}$ and $L_{ins}^{PEBAL}$, we obtain the overall prediction-based objective $L_{pred}^{PEBAL}$ by:

$$L_{pred}^{PEBAL} = \alpha L_{glob}^{PEBAL} + (1 - \alpha) L_{ins}^{PEBAL} \tag{45}$$

### B.1.3 LNOIB on Mask2Anomaly

M2A is based on the segmentation model Mask2Former [8], which decouples the prediction of masks and categories. Despite this difference, it ultimately yields a pixel-wise prediction $f(x) \in \mathbb{R}^{K \times H \times W}$ as well. Note that, $f(x)$ is the logit here, rather than softmax prediction.

According to [41], $l_{in}^{M2A}$ and $l_{out}^{M2A}$ are calculated by:

$$\begin{cases} l_{in}^{M2A} = \frac{1}{2} (\max_{c=1}^{K} f_c(x_i))^2 \\ l_{out}^{M2A} = \frac{1}{2} \left( \max(0, \gamma + \max_{c=1}^{K} f_c(x_i)) \right)^2 \end{cases} \tag{46}$$

where $\gamma$ is a hyperparameter that decides the minimum distance between ID and OOD classes. Minimizing $l_{in}^{M2A}$ and $l_{out}^{M2A}$ aims to create a gap between ID and OOD logits, which also makes the anomaly score closer to the binary ground-truth (normality or anomaly), as demonstrated in Eq. (36). Please refer to [41] for detailed information.

Then the global-level objective for M2A is calculated as:

$$L_{glob}^{M2A} = \frac{1}{N_{in}} \sum_{x_i \in \Omega_{in}} l_{in}^{M2A} + \frac{1}{N_{out}} \sum_{x_j \in \Omega_{out}} l_{out}^{M2A} \tag{47}$$

Building on $l_{in}^{M2A}$ and $l_{out}^{M2A}$, our instance-level objective adapted to M2A can be formulated as:

$$L_{ins}^{M2A} = \frac{1}{N_{in}} \sum_{x_i \in \Omega_{in}} l_{in}^{M2A} + \frac{1}{N} \sum_{o_k \in O} \sum_{x_j \in \Omega_{o_k}} \frac{1}{|\Omega_{o_k}|} \cdot l_{out}^{M2A} \tag{48}$$

Then the prediction-based objective for M2A can be formulated as:

$$L_{pred}^{M2A} = \alpha L_{glob}^{M2A} + (1 - \alpha) L_{ins}^{M2A} \tag{49}$$

### B.1.4 Summary

Above, we demonstrate how the instance-level objective in LNOIB adapts to current OOD fine-tuning strategies, including EM, PEBAL, and M2A. $L_{ins}$ is not a specific loss function; instead, $L_{ins}$ can be built upon each $l_{in}$ and $l_{out}$ and thus can be technically adapted to existing OOD fine-tuning strategies that employ global-level objectives, showcasing the versatility of LNOIB.

### B.2 Feature Separation Objective

This objective can also be easily integrated into current OOD fine-tuning strategies. Since each OOD fine-tuning strategy is based on a semantic segmentation model, we can leverage the feature map from the backbone to generate prototypes and calculate the feature separation objectives. Specifically, EM and PEBAL use DeepLabv3+ [6] with the WideResNet38 [56] backbone, while M2A employs Mask2Former [8] with global masked attention [41] with the ResNet50 [20] backbone. Accordingly, we use the feature maps from stages 2, 3, and 4 of each backbone, resize them to 1/4 of the original input image size, and add these features together for prototype establishment.

### B.3 Configurations of OOD Fine-Tuning

We apply LNOIB to current OOD fine-tuning strategies, including EM, PEBAL, and M2A. Apart from altering the overall objective, we follow the configurations used in each respective strategy without

bells and whistles, including preprocessing methods, training epochs, learning rates, optimizers, batch sizes, random seeds, and other settings. Our aim is to present the versatility of LNOIB and validate whether LNOIB can enhance existing OOD fine-tuning strategies, although we acknowledge that further adjusting these settings could potentially lead to higher performance for each respective approach. For the detailed configurations, please refer to the official github repos of EM[2], PEBAL[3], and M2A[4]. For each experiment, we conduct 3 times and report the average performance to guarantee a fair comparison.

As to the hardware, we use a server running Ubuntu 22.04, equipped with 4 RTX 3090Ti GPUs, each with 24 GB of memory, as well as another server with 2 NVIDIA A100 GPUs, each with 80 GB of memory. We adopt the Pytorch framework to conduct the training and evaluation process.

## C  Component-Level Metrics

The component-level evaluation metrics are introduced in [4], which are designed to focus on detecting anomalies, regardless of their sizes. These metrics are essential because pixel-level metrics may not adequately penalize a model for missing small anomalies, even though such anomalies might be critical to detect. Evaluating this metric is crucial for LNOIB, as our target is to capture all anomalies using such an OOD fine-tuning strategy. For a comprehensive component-level assessment of detected anomalies, we need to consider component-wise true positives (*TP*), false negatives (*FN*), and false positives (*FP*). These quantities are measured by treating anomalies as the positive class. From these measurements, we are able to use three metrics, i.e., sIoU, PPV, and F1*, to evaluate component-wise segmentation performance of anomalies. Below, we detail the computation of these metrics, using $\mathcal{O} = \{\imath_1, \imath_2, ..., \imath_T\}$ to represent the set of ground-truth components and $C = \{c_1, c_2, ..., c_N\}$ to denote the set of predicted components.

**sIoU** employed in SMIYC [3] is a modified version of the component-wise intersection over union proposed in [43]. It mainly considers the ground-truth components in the computation of *TP* and *FN*. The sIoU score for a ground-truth component $\imath_k$ can be formulated as:

$$\text{sIoU}(\imath_t) = \frac{|\imath_t \cap C(\imath_t)|}{|(\imath_t \cup C(\imath_t)) \setminus \mathcal{A}(\imath_t)|}, \quad C(\imath_k) = \bigcup_{c_k \in C, c_k \cap \imath_t \neq \emptyset} c_k \tag{50}$$

where $\mathcal{A}(\imath_t)$ is a term that excludes from the union of those pixels that correctly intersect with other ground-truth components different from $\imath_t$. $C(\imath_t)$ represents the set of predicted components that intersect with $\imath_t$.

Accordingly, given a threshold $\eta \in [0, 1]$, a target $\imath_t \in \mathcal{O}$ is considered as a *TP* if $sIoU(\imath_t) > \eta$, otherwise an *FN*.

**PPV** measures whether a predicted component $c_k \in C$ belongs to *FP*, and it is formulated as:

$$\text{PPV}(c_k) = \frac{|c_k \cap \mathcal{O}(c_k)|}{|c_k|}, \quad \mathcal{O}(c_k) = \bigcup_{\imath_k \in \mathcal{O}, \imath_t \cap c_k \neq \emptyset} \imath_t \tag{51}$$

where $\mathcal{O}(c_k)$ represents the set of ground-truth components that intersect with the predicted component $c_k$. A predicted component $c_k$ is an FP if $PPV(c_k) \leq \eta$.

**F1*** is calculated based on the results of sIoU and PPV by:

$$F_1 * (\eta) = \frac{2TP(\eta)}{2TP(\eta) + FN(\eta) + FP(\eta)} \in [0, 1] \tag{52}$$

where *TP*, *FN*, and *FP* are determined by the value of $\eta$.

---

[2]https://github.com/robin-chan/meta-ood
[3]https://github.com/tianyu0207/PEBAL
[4]https://github.com/shyam671/Mask2Anomaly-Unmasking-Anomalies-in-Road-Scene-Segmentation

Table 4: Close-set semantic segmentation results on Cityscapes, which are presented in three groups of experiments. In each group, the first line shows the performance of the vanilla semantic segmentation model. The second line presents the results after OOD fine-tuning with current strategies (EM, PEBAL, and M2A) based on the segmentation model. The third line represents the performance after OOD fine-tuning with LNOIB based on each OOD fine-tuning strategy mentioned in the previous line.

| Approach | mIoU ↑ |
|---|---|
| DeepLabv3+ (WideResNet38) | 77.85 |
| +EM | 76.71 |
| +LNOIB (based on EM) | 77.33 |
| DeepLabv3+ (WideResNet38) | 77.85 |
| +PEBAL | 77.21 |
| +LNOIB (based on PEBAL) | 77.14 |
| Mask2Former (ResNet50) | 80.14 |
| +M2A | 79.39 |
| +LNOIB (based on M2A) | 79.18 |

These component-level metrics assess component locations independently of their sizes, ensuring that larger components do not dominate the metrics. Following SMIYC, we set $\eta = 0.5$ for evaluation.

Besides SMIYC benchmark, we also employ the component-level metrics to explore the coverage of small anomalies on Road Anomaly dataset and the validation sets of FS Static and FS L&F (as the test set of Fishyscapes benchmark does not provide an API to evaluate such component-level metrics). Accordingly, in Table 1, we reproduce the component-level results of previous approaches on SMIYC RA, SMIYC RO, Road Anomaly, and the validation sets of FS Static and FS L&F, respectively. This enables a fair evaluation of how our instance-level extension improves the performance of each baseline. If we were to directly adopt the benchmark results reported in the original papers, the comparison would be less accurate due to minor discrepancies between their setups (such as hardware discrepancy and random seed discrepancy, as we report the mean of 3 runs) and our reproduced baselines. Although these differences are relatively small, using our own reproduced results allows for a more precise assessment of the performance gains brought by our instance-level extension. Based on the experimental results, we find that the combination of LNOIB and current OOD-fine-tuning strategies significantly enhances the component-level metrics, and the incorporation of M2A [42] and LNOIB achieves the top-performing sIoU, PPV, and F1* results in most cases.

## D   Close-set Segmentation Performance

OOD fine-tuning can potentially reduce the performance of the close-set semantic segmentation. To investigate this effect, we evaluate the mean Intersection over Union (mIoU) on the Cityscapes dataset. We conduct three groups of experiments: in each group, we compare the results of the vanilla segmentation model, the model fine-tuned with current global-level objectives, and the model fine-tuned with LNOIB (built on each global-level objective). The results of this comparison are presented in Table 4. We observe that adopting LNOIB results in only a 0.52% to 0.96% mIoU drop compared to the vanilla segmentation model. In most cases, the mIoU results are slightly higher than those obtained through existing OOD fine-tuning with global-level objectives.

## E   More Ablation Results

### E.1   Fine-grained Balance of $L_{pred}$

In the main paper, we analyze the effects of combining $L_{glob}$ and $L_{ins}$, and observe that their joint use yields superior performance. In this section, we further investigate the impact of the weighting factor $\alpha$ with respect to the performance. The results are depicted in Figure 5 (a)-(c). Interestingly, $\alpha = 0.5$ achieves the top-performing results in most cases. We observe that when $\alpha \leq 0.4$, there is a significant decline in pixel-level performance (AuPRC in red). Conversely, when $\alpha \geq 0.6$, the improvements in

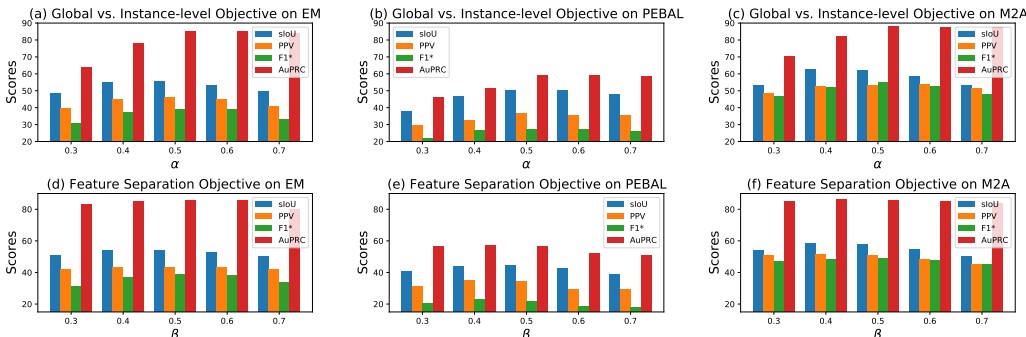

Figure 5: Results for varying balanced factors, i.e., $\alpha$ in $L_{pred}$ (first row) and $\beta$ in $L_{feat}$ (second row). The settings of $\alpha = 0.5$ and $\beta = 0.5$ achieve competitive performances, although they are not the best across all metrics for all OOD fine-tuning strategies.

component-level metrics (sIoU, PPV, and F1*) diminish. Therefore, as a compromise, $\alpha = 0.5$ offers competitive results across all OOD fine-tuning strategies, including EM, PEBAL, and M2A.

## E.2 Fine-grained Balance of $L_{feat}$

In the main paper, we explore the effects of incorporating $L_{sem}$ and $L_{near}$, and find that their joint use yields superior performance. Here, we aim to further identify the optimal balance factor $\beta$, as illustrated in Figure 5 (d)-(f). Our findings indicate that $\beta = 0.5$ delivers competitive performance across EM, PEBAL, and M2A. For simplicity and versatility, we adopt $\beta = 0.5$ in the final results, despite the real optimum potentially varying for each OOD fine-tuning strategy. To achieve the optimal performance for specific approaches, we can further explore fine-grained intervals within $[0.4, 0.6]$.

## E.3 Balance of LNOIB Objective

In our main paper, we formulate the overall LNOIB objective as:

$$L_{LNOIB} = \gamma_1 L_{pred} + \gamma_2 L_{feat} \tag{53}$$

In $L_{LNOIB}$, we simply set $\gamma_1 = \gamma_2 = 1$. This decision is based on the following considerations.

Firstly, the ranges of $L_{glob}$ and $L_{ins}$ in $L_{pred}$ are the same, both being relevant to the definitions of $l_{in}$ and $l_{out}$. Moreover, the ranges of $L_{sem}$ and $L_{near}$ in $L_{feat}$ are also identical, both falling between 0 and 1. However, the ranges of $L_{pred}$ and $L_{feat}$ are completely different, making it challenging to set a uniform balanced factor.

Although setting proper balance factors might further improve the overall performance, the search space will be significantly larger compared to $L_{pred}$ and $L_{feat}$. Furthermore, since the ranges of $l_{in}$ and $l_{out}$ vary across different global-level objectives, whether a balance factor is adaptable to all existing strategies needs further consideration.

In addition, the range of $L_{feat}$ is $[0, 1]$, and $L_{pred}$ is in the same order of magnitude as $L_{feat}$ and typically larger than $L_{feat}$, indicating that $L_{feat}$ acts as an auxiliary objective to enhance OOD fine-tuning, which is consistent with our experimental findings in Table 3 of the main paper, and Figure 5 of the supplementary material. Therefore, we first adopt the ostrich strategy that sets $\gamma_1 = \gamma_2 = 1$ here. Interestingly, this setting is well adapted to various OOD fine-tuning strategies including EM, PEBAL, and M2A.

We then briefly explore the optimal values of the balancing factors $\gamma_1$ and $\gamma_2$. Specifically, we fix $\gamma_1 = 1$ and vary $\gamma_2 \in \{0.01, 0.1, 1, 10, 100\}$. The corresponding results are shown in Figure 6. We observe that setting $\gamma_2 = 1$ yields the best performance. A larger $\gamma_2$ reduces the influence of the prediction-based objective, leading to a noticeable performance degradation. Specifically, when $\gamma_2 = 100$, the impact of $L_{pred}$ becomes negligible. As a result, the direct objective for OOD

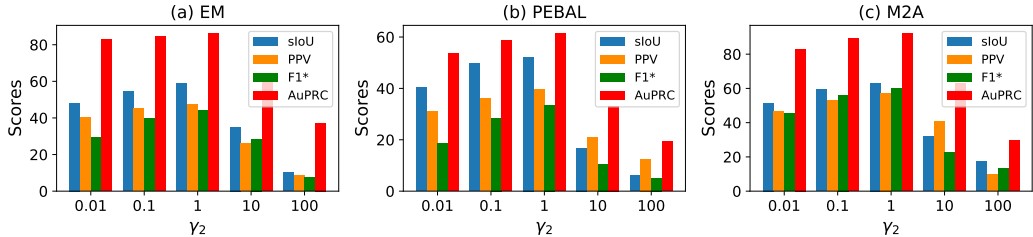

Figure 6: Results for varying balanced factors $\gamma_1$ and $\gamma_2$. We fix $\gamma_1 = 1$ and search the best value of $\gamma_2 \in \{0.01, 0.1, 1, 10, 100\}$. The setting of $\gamma_1 = \gamma_2 = 1$ achieves competitive performances in most cases.

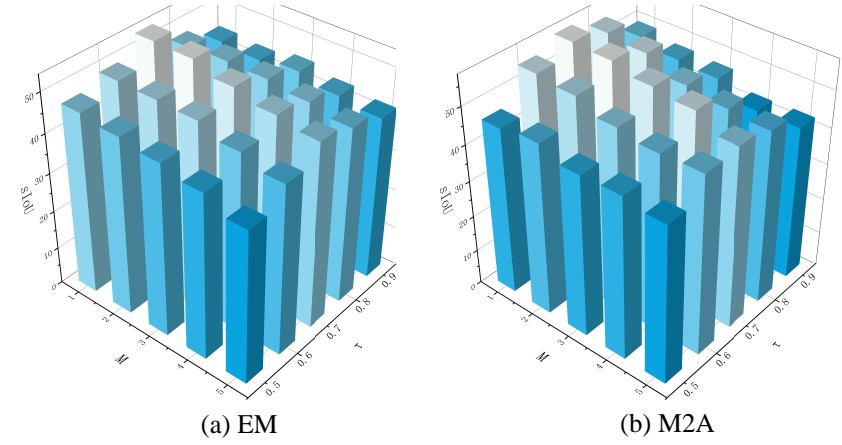

Figure 7: sIoU results for different combinations of nearest number $M$ and threshold $\tau$ on SMIYC-RA. The combination of $M = 1$ and $\tau = 0.7$ yields superior results.

fine-tuning on the predicted anomaly scores is largely diminished. Conversely, a smaller $\gamma_2$ (such as $\gamma_2 = 0.01$) results in suboptimal outcomes, as the feature-level optimization also plays a crucial role in effectively separating ID and OOD representations. According to the results above, we finally select $\gamma_1 = \gamma_2 = 1$ as the balancing factors in overall objectives, as this setting performs well on all of these approaches.

## E.4 Thresholds of $\tau$ and $M$

Additionally, we investigate the impact of the threshold $\tau$ and the number of nearest neighbors $M$ for $L_{feat}$. We evaluate the sIoU results on EM and M2A, as shown in Figure 7. The figure shows that $\tau = 0.7$ achieves the best performance. We believe that a smaller $\tau$ introduces noise in constructing ID prototypes, while a larger $\tau$ results in incomplete ID prototypes. Furthermore, we find that $M = 1$ already yields competitive results in most cases.

## E.5 Ablation Study on More Datasets

In the main paper and supplementary materials above, we mainly conduct ablation study on SMIYC-RA. To further evaluate the generality of each component, we conduct more ablation study using M2A on FS L&F and Road Anomaly datasets.

Firstly, we explore the combination of $L_{glob}$ and $L_{ins}$ in prediction-based objective, and the results are demonstrated in Table 6. Moreover, we also investigate the combination of $L_{sem}$ and $L_{near}$ in

Table 5: Instance-level results on the validation set FS Static and FS L&F. By incorporating LNOIB, existing OOD fine-tuning strategies achieve higher performances. The results in **bold** represent better performances compared with vanilla approaches. ↑ means higher values are better.

| Approach | FS Static | | FS L&F | |
|---|---|---|---|---|
| | iAP ↑ | iAP50 ↑ | iAP ↑ | iAP50 ↑ |
| EM (ICCV'21) | 23.7 | 30.1 | 25.8 | 37.5 |
| **+LNOIB(ours)** | **33.4** | **37.1** | **33.8** | **47.9** |
| PEBAL (ECCV'22) | 23.8 | 32.1 | 19.6 | 25.3 |
| **+LNOIB(ours)** | **29.4** | **40.7** | **26.8** | **37.2** |
| M2A (PAMI'24) | 26.8 | 41.0 | 27.3 | 36.9 |
| **+LNOIB(ours)** | **37.6** | **49.3** | **34.0** | **47.3** |

Table 6: Ablation results of each component in prediction-based objective for M2A on more FS L&F and Road Anomaly. Incorporating $L_{ins}$ contributes to further improvements.

(a) FS L&F

| $L_{glob}$ | $L_{ins}$ | sIoU ↑ | PPV ↑ | F1* ↑ | AuPRC ↑ | FPR$_{95}$ ↓ |
|---|---|---|---|---|---|---|
| ✓ | | 26.51 | 19.43 | 23.76 | 89.91 | 1.85 |
| | ✓ | 25.18 | 15.69 | 17.22 | 57.30 | 8.97 |
| ✓ | ✓ | **37.58** | **35.10** | **34.23** | **90.38** | **1.77** |

(b) Road Anomaly

| $L_{glob}$ | $L_{ins}$ | sIoU ↑ | PPV ↑ | F1* ↑ | AuPRC ↑ | FPR$_{95}$ ↓ |
|---|---|---|---|---|---|---|
| ✓ | | 47.43 | 40.80 | 44.57 | 75.70 | 16.31 |
| | ✓ | 35.80 | 31.68 | 37.39 | 53.04 | 21.54 |
| ✓ | ✓ | **48.99** | **50.14** | **46.73** | **78.01** | **14.27** |

feature separation objective, and the results are presented in Table 7. Both tables demonstrate the superiority of $L_{ins}$, $L_{sem}$, and $L_{near}$ in our instance-level OOD fine-tuning framework, which is consistent to the results of SMIYC-RA.

### E.6 OOD Fine-tuning with Imperfect Instance Masks

In the main experiments, we adopt a copy-and-paste strategy using ground-truth masks for OOD fine-tuning. However, when anomaly masks are unavailable, we should rely on predicted masks generated by several instance segmentation models, which are inevitably imperfect.

In this part, we try to explore how sensitive is performance to imperfect instance masks. To simulate imperfect instance masks, we replace the GT masks in COCO with the predicted masks from Mask R-CNN [19] (34.7 AP), which tend to have coarse boundaries. Using these masks, we apply the same copy-paste strategy to create mix-content images. Experimental results of M2A on FS L&F and Road Anomaly are presented in Table 8, showing that while using imperfect masks leads to a slight performance drop compared to GT masks, our OOD fine-tuning method remains effective. We attribute this to the fact that the fine-tuning mainly captures the overall OOD patterns. Imperfect masks may introduce noise into the OOD instance-level prototypes and thus slightly weaken the feature separation objective. But the impact is limited, showcasing the robustness of our method.

## F  Anomaly Instance Segmentation

The SMIYC and Fishyscapes benchmarks, designed specifically for anomaly semantic segmentation, do not inherently support distinguishing between different anomalies. Common approaches such as EM, PEBAL, and M2A are similarly limited in this regard. However, the instance-level metrics proposed in [37] provide a means to evaluate whether LNOIB can enhance segmentation performance, particularly for small anomalies. These metrics are employed to further confirm that LNOIB leads to tangible improvements at the instance level.

Table 7: Ablation results of each component in feature separation objective for M2A on more FS L&F and Road Anomaly. Both $L_{sem}$ and $L_{near}$ contribute to further improvements in $L_{feat}$.

(a) FS L&F

| $L_{sem}$ | $L_{near}$ | sIoU ↑ | PPV ↑ | F1* ↑ | AuPRC ↑ | FPR$_{95}$ ↓ |
|---|---|---|---|---|---|---|
| | | 26.51 | 19.43 | 23.76 | 89.91 | 1.85 |
| ✓ | | 28.11 | 22.04 | 24.46 | 89.33 | 1.91 |
| | ✓ | 31.58 | 28.40 | 27.81 | 89.74 | 2.03 |
| ✓ | ✓ | **35.85** | **33.93** | **32.10** | **90.07** | **1.72** |

(b) Road Anomaly

| $L_{sem}$ | $L_{near}$ | sIoU ↑ | PPV ↑ | F1* ↑ | AuPRC ↑ | FPR$_{95}$ ↓ |
|---|---|---|---|---|---|---|
| | | 47.43 | 40.80 | 44.57 | 75.70 | 16.31 |
| ✓ | | 45.39 | 41.17 | 43.08 | 73.36 | 17.21 |
| | ✓ | 47.93 | 44.57 | 45.61 | 76.44 | 16.13 |
| ✓ | ✓ | **50.51** | **48.87** | **50.05** | **78.26** | **15.45** |

Table 8: Results of OOD fine-tuning with imperfect masks generated by Mask-RCNN.

| Mask type | FS L&F | | | | | Road Anomaly | | | | |
|---|---|---|---|---|---|---|---|---|---|---|
| | sIoU | PPV | F1* | AuPRC | FPR$_{95}$ | sIoU | PPV | F1* | AuPRC | FPR$_{95}$ |
| GT | 39.16 | 38.10 | 34.80 | 90.54 | 1.68 | 53.70 | 51.46 | 50.97 | 75.70 | 16.31 |
| Mask-RCNN | 38.39 | 36.60 | 33.86 | 88.73 | 1.91 | 50.58 | 51.07 | 47.92 | 75.37 | 16.70 |

## F.1 Instance-level Metrics and Results

In addition to the official component-level metrics, we also evaluate the instance-level metrics of LNOIB on the Fishyscapes benchmark [1], following the setup in [37]. These metrics include instance-level average precision (iAP) and iAP50. Since previous approaches do not utilize instance-level metrics, we primarily compare the performance of vanilla EM, PEBAL, and M2A with their counterparts enhanced by LNOIB. As shown in Table 5, incorporating LNOIB significantly improves iAP and iAP50 for OOD fine-tuning approaches, indicating enhanced performance on small anomalies. Since small and large anomalies occupy similar proportions, and large anomalies are already well segmented, these improvements likely highlight the enhanced segmentation of smaller anomalies.

## F.2 Instance Anomaly Segmentation Details

To compute these instance-level metrics, we utilize the anomaly semantic segmentation results along with the ground-truth instance masks to estimate the corresponding predicted instance masks, following the setup in [37]. Specifically, given the predicted anomaly semantic segmentation result $\hat{y} \in \{0, 1\}^{H \times W}$, we first obtain the anomaly pixel set $\Omega_A$, where pixels with a value of 1 indicate an anomaly. Then, based on the ground-truth instance mask $\Omega_{o_k}$ of the $k$-th anomaly, we find the pixel set $P = \{p_1, p_2, ..., p_{N_p}\} \subset \Omega_A$, which satisfies the following properties: 1) $\forall p \in P$, s.t. $p \in \Omega_{o_k}$; 2) $\nexists p \in \Omega_A \backslash P$, s.t. $p \in \Omega_{o_k}$. Accordingly, the estimated instance mask of $k$-th anomaly can be represented as $\hat{o_k} = \{c_1, c_2, ..., c_{N_p}\}$, where $c_i$ denotes the connected component associated with pixel $p_i$. Using this representation, we can derive the anomaly instance segmentation results and subsequently compute the iAP and iAP50 metrics.

Note that, iAP and iAP50 are equivalent to AP and AP50 in Cityscapes [9], with the key difference being that the former evaluates the average precision of anomalous instances. Similar to [37], as SMIYC validation sets do not provide instance masks, we evaluate the instance-level metrics on FS Static and FS L&F datasets.

## G  More Clarification of Instance-level OOD Fine-tuning and Anomaly Instance Segmentation

The core idea of LNOIB is to leverage instance-level OOD fine-tuning to enhance anomaly segmentation. Since OOD fine-tuning requires a mixed-content dataset, we use the widely adopted AnomalyMix, which pastes COCO instances onto Cityscapes images. Previous OOD fine-tuning methods do not consider instance-level contrastive loss and thus do not need to differentiate between different anomalies when preparing the mixed-content dataset. In contrast, we assign unique instance labels to different anomalies, enabling effective instance-level OOD fine-tuning. Note that, the instance-aware annotation is simple and straightforward in this copy-and-paste approach to generate the mixed-content dataset.

Unless explicitly stated otherwise, anomaly segmentation generally refers to anomaly semantic segmentation. SMIYC [3] and Fishyscapes [1], and Road Anomaly datasets all focus on this task,

which involves segmenting anomalous regions. The component-level metrics assess how well the anomalous components are covered, giving greater weight to small components compared to pixel-level metrics.

As our main claim is that LNOIB improves performance on small anomalies, evaluating instance-level metrics [37] provides further validation of this claim. Therefore, we also assess it using the anomaly instance segmentation task. While methods on SMIYC and Fishyscapes, such as EM, PEBAL, and M2A, focus on anomaly semantic segmentation, we adopt the strategy outlined in Section F of the appendix to simulate anomaly instance segmentation results.

Overall, both the component-level metrics in anomaly semantic segmentation and the instance-level metrics in anomaly instance segmentation demonstrate that OOD fine-tuning with LNOIB improves the segmentation quality of small anomalies.

