# OpenReview forum: "Leaving No OOD Instance Behind: Instance-Level OOD Fine-Tuning for Anomaly Segmentation"
_NeurIPS.cc/2025/Conference — NeurIPS 2025 poster_

### Official Review · Reviewer_hxoy · 2025-07-01

**Clarity:** 2
**Significance:** 2
**Originality:** 3
**Rating:** 4
**Confidence:** 4

**Summary:**

The paper targets anomaly (OOD) segmentation in autonomous-driving datasets and argues that existing OOD fine-tuning methods optimize global pixel-wise objectives, thereby overlooking small OOD instances. The authors (1) provide a theoretical analysis showing that a dominant large anomaly can mask gradient signal for small ones, (2) propose adding an “instance-level objective” Lins that averages the loss over each OOD instance rather than over all pixels, and (3) introduce a feature-separation loss. The combined framework LNOIB is positioned as a plug-in for three existing fine-tuning methods (Entropy Maximization, PEBAL, Mask2Anomaly).

**Questions:**

- How sensitive is performance to imperfect instance masks?

- Can LNOIB operate when anomaly masks are unavailable?

- Provide mean, std over the 3 runs.

**Ethical Concerns:**

["NO or VERY MINOR ethics concerns only"]

**Final Justification:**

Thanks for the authors' response. I believe most of my concerns are addressed. However, I would encourage the authors to state clearly why reproduced results are used in the experiment section to enhance reproducibility. Also, the results of "imperfect instance masks" shall be presented in the paper, which would significantly enhance the proposed methods' applicability.

I maintain my score as BA

**Limitations:**

See weaknesses and questions.

**Quality:**

3

**Strengths And Weaknesses:**

Strengths

- Well-motivated focus on small anomalies. Simple, generic objectives that can be inserted into multiple existing pipelines.

- Extensive experiments on five datasets.

- No additional inference cost and tiny drop on ID classes.

Weaknesses

- Novelty is incremental. Instance-wise re-weighting and prototype separation are known ideas in few-shot segmentation and object re-ID; paper does not clearly situate itself w.r.t. those.

- The experiments are reproduced by authors. Why do not follow the benchmark by original papers like M2A?

- The bounds in the theoretical part ignore feature-separation loss.

- Authors run 3 seeds but only averages are shown; no std/CI or p-values for the headline tables.

- Construction of “OOD instances” relies on copy-paste from COCO with instance labels. Can LNOIB operate when anomaly masks are unavailable?

---

> ### Author Rebuttal · Authors · 2025-07-25
>
> We sincerely appreciate your encouraging feedback and helpful suggestions. We will give a detailed response to each of your insightful concerns below.
>
> **W1. Novelty is incremental. Instance-wise re-weighting and prototype separation are known ideas in few-shot segmentation and object re-ID; paper does not clearly situate itself w.r.t. those.**
>
> Thank you for your valuable comment! While the use of prototypes has been explored in few-shot segmentation (FSS) and object re-identification (re-ID), our method is fundamentally different from them: 1) In our work, OOD prototypes are constructed based on each individual OOD instance, rather than representing an ID semantic category (in FSS) or a specific ID object (in re-ID). This construction of OOD prototype is a fundamental innovation tailored to the unique nature of OOD detection. 2) We provide a novel theoretical foundation of why using OOD prototypes enhance small anomaly segmentation in the feature separation loss, which is unique in anomaly segmentation. Please see the response to W3. 3) In FSS and re-ID tasks, prototypes are directly used to match query image pixels for classification [1]. In contrast, our method formulates prototype usage as an optimization problem, where the key objective is to explicitly push the OOD prototypes away from the ID semantic prototypes. Although [2] in FSS aims to separate ID features, its goals and challenges are different from those in our OOD task. Our work is the first to explore the separation between OOD instance prototypes and ID category prototypes in anomaly segmentation. In summary, the prototype construction, the theoretical foundation, and the target of our feature separation strategy differ fundamentally from the work in FSS and re-ID.
>
> As to the instance-wise re-weighting strategy, this idea naturally arises from the intuition that critical small anomalies deserve more focus. To our knowledge, this is the first work that explicitly targets small anomalies at instance-level in anomaly segmentation, supported by our rigorous theoretical foundation. We identify the limitations of previous global-level objectives and propose an extensive instance-level paradigm, not tied to any particular loss function. This is the first instance-based fine-tuning strategy designed for anomaly segmentation and it yields remarkable progress in segmenting small anomalies.
>
> Following your helpful comment, we will include a new subsection in Related Work to more thoroughly compare our approach with prior FSS and re-ID methods, especially in terms of prototype construction, optimization objectives, and theoretical foundations of our instance-level design for both objectives.
>
> **W2. The experiments are reproduced by authors. Why do not follow the benchmark by original papers like M2A?**
>
> Thank you for pointing out this issue! As our core contribution lies in enhancing existing OOD fine-tuning methods with our instance-level objective, we believe it is reasonable to compare the results with and without this extension under the same experimental settings. Importantly, these settings strictly follow those adopted in prior works, rather than being designed by ourselves. We also note that some of the reproduced baseline results even outperform those originally reported, further confirming the robustness of our implementation.
>
> **W3. The bounds in the theoretical part ignore feature-separation loss.**
>
> Thank you for pointing out this issue! Below, we provide a theoretical analysis of the feature separation loss when *globally separating each OOD pixel-wise representation from ID prototypes*. This analysis shows that, without instance-level OOD prototypes, small anomalies are likely to be ignored in the feature separation loss.
>
> $$L_{gsem}=\frac{1}{K|\Omega_{out}|} \sum_{q_{c}\in Q} \sum_{x_{j}\in \Omega_{out}}cosSim(F_{x_{j}},q_{c})$$
>
> **Lemma 2.** The total loss of $L_{gsem}$ decomposes into instance-specific components:
>
> $$L_{gsem}=\frac{1}{K|\Omega_{out}|} \sum_{q_{c}\in Q} \sum_{o_{k}\in O} |\Omega_{o_{k}}|\cdot {\mathbb{E}}_{x_j \sim \Omega\_{o\_{k}}} cosSim(F\_{x_j},q_c) \triangleq \sum\_{k=1}^{N} r\_{k}L\_{gsem,o\_{k}}$$
>
> where $r_{k}=\frac{|\Omega_{o_{k}}|}{|\Omega_{out}|}$ is the normalized weight of instance, and $L\_{gsem,o\_{k}}=\frac{1}{K} \mathbb{E}\_{x\_j \sim \Omega\_{o\_{k}}} \sum\_{q\_{c} \in Q}  cosSim(F\_{x_j},q\_c)$ is the mean ID semantic loss over $\Omega\_{o\_{k}}$.
>
> Proof:
>
> $$L_{gsem}=\frac{1}{K|\Omega_{out}|} \sum_{q_{c}\in Q} \sum_{x_{j}\in \Omega_{out}}cosSim(F_{x_{j}},q_{c})=\frac{1}{K|\Omega_{out}|} \sum_{q_{c}\in Q} \sum_{o_{k}\in O}\sum_{x_{j}\in o_{k}}cosSim(F_{x_{j}},q_{c})=\frac{1}{K|\Omega_{out}|} \sum_{q_{c}\in Q} \sum_{o_{k}\in O} |\Omega_{o_{k}}|\cdot {\mathbb{E}}_{x_j \sim \Omega\_{o\_{k}}} cosSim(F\_{x_j},q_c)$$
>
> Based on the definition of $L_{gsem,o_{k}}$, we have:
>
> $$L_{gsem}=\frac{1}{|\Omega_{out}|} \sum_{o_{k}\in O} |\Omega_{o_k}| \cdot L_{gsem,o_k} \quad\square$$
>
> **Theorem 3.** If there exists $t\in\{1,2,...,N\}$, such that $|\Omega_{t}|\ge (1-\epsilon)|\Omega_{out}|$ for some $\epsilon \in(0,1)$, then $L\_{gsem}$ is bounded by $(1-\epsilon)L\_{gsem,o\_{t}}\le L\_{gsem} \le (1-\epsilon)L\_{gsem,o\_{t}}+\epsilon \cdot \max\_{k\ne t} L\_{gsem,o\_{k}}$.
>
> Proof: The proof follows a reasoning analogous to that of Theorem 1.
>
> **Corollary 2.** If $\frac{|\Omega_{t}|}{|\Omega_{out}|}\rightarrow 1$, then $L_{gsem}\rightarrow L_{gsem,o_{t}}$ in probability.
>
> Proof: This proof mirrors the approach taken in Corollary 1.
>
> Based on the theoretical analysis above, globally separating each OOD feature from ID prototypes also overlooks the optimization for small anomalies, thus our ID semantic objective $L_{sem}$ in Eq. (9) is more reasonable, which equally optimizes each anomaly. Moreover, the theory of using $L_{near}$ is the same.
>
> **W4&Q3. Authors run 3 seeds but only averages are shown; no std/CI or p-values for the headline tables.**
>
> Thank you for this great suggestion! We provide the mean and std of our results in Tables 1 and 2 below.
>
> Results on component-level metrics:
> | | | SMIYC-RA| | |SMIYC-RO| | |FS-Static| | |FS L\&F| | |Road Anomaly| |
> |-|:-:|:-:|:-:|:-:|:-:|:-:|:-:|:-:|:-:|:-:|:-:|:-:|:-:|:-:|:-:|
> |**Method**|**sIoU**|**PPV**|**F1***|**sIoU**|**PPV**|**F1***|**sIoU**|**PPV**|**F1***|**sIoU**|**PPV**|**F1***|**sIoU**|**PPV**|**F1***|
> |EM+LNOIB|58.73|47.42|43.90|50.86|58.80|48.57|44.38|37.20|37.79|34.74|29.56|23.97|35.72|33.18|27.30|
> | |(±0.26)|(±0.21)|(±0.35)|(±0.38)|(±0.22)|(±0.09)|(±0.30)|(±0.41)|(±0.25)|(±0.17)|(±0.24)|(±0.10)|(±0.18)|(±0.05)|(±0.29)
> |PEBAL+LNOIB|52.17|39.55|33.48|32.88|12.90|13.46|38.84|29.75|28.44|27.36|35.78|17.73|37.68|30.91|32.65|
> | |(±0.41)|(±0.13)|(±0.44)|(±0.30)|(±0.06)|(±0.16)|(±0.25)|(±0.19)|(±0.09)|(±0.26)|(±0.33)|(±0.14)|(±0.33)|(±0.15)|(±0.12)
> |M2A+LNOIB|63.15|57.37|60.08|61.50|73.71|70.25|52.08|44.93|42.96|39.16|38.10|34.80|53.70|51.46|50.97|
> | |(±0.48)|(±0.53)|(±0.27)|(±0.07)|(±0.51)|(±0.46)|(±0.18)|(±0.20)|(±0.30)|(±0.31)|(±0.17)|(±0.19)|(±0.38)|(±0.25)|(±0.08)
>
> Results on pixel-level metrics:
> | |SMIYC-RA| |SMIYC-RO| |FS-Static| |FS L\&F| |Road Anomaly| |
> |-|:-:|:-:|:-:|:-:|:-:|:-:|:-:|:-:|:-:|:-:|
> |**Method**|**AuPRC**|**FPR$_{95}$**|**AuPRC**|**FPR$_{95}$**|**AuPRC**|**FPR$_{95}$**|**AuPRC**|**FPR$_{95}$**|**AuPRC**|**FPR$_{95}$**|
> |EM+LNOIB|86.26|15.28|84.47|2.39|86.95|8.89|81.68|5.18|72.42|19.74
> | |(±0.47)|(±0.11)|(±0.35)|(±0.06)|(±0.21)|(±0.10)|(±0.24)|(±0.07)|(±0.68)|(±0.17)
> |PEBAL+LNOIB|61.46|22.03|31.30|18.44|83.81|4.47|64.50|3.82|65.35|22.77
> | |(±0.44)|(±0.18)|(±0.07)|(±0.14)|(±0.52)|(±0.03)|(±0.40)|(±0.11)|(±0.14)|(±0.10)
> |M2A+LNOIB|92.20|11.34|92.58|0.33|75.19|8.27|90.54|1.68|79.38|13.29
> | |(±0.11)|(±0.07)|(±0.42)|(±0.04)|(±0.55)|(±0.11)|(±0.58)|(±0.05)|(±0.22)|(±0.08)
>
> Based on the mean and std across three runs, we find that the std remains consistently low—typically within 1% of the mean—indicating that our method exhibits stable performance across different runs.
>
> **W5&Q2. Can LNOIB operate when anomaly masks are unavailable?**
>
> Thank you for raising this important point! LNOIB cannot function without anomaly masks. This is because the instance-level objective in Eq. (5) relies on the presence of anomaly masks to identify OOD instances. Similarly, the feature separation loss in Eq. (13) depends on OOD prototypes, which are constructed based on the anomaly masks in Eq. (7). Please note that, the copy-paste method is easy and common in previous OOD fine-tuning methods to generate mix-content images, while they treat all anomalies as the whole OOD region.
>
> **Q1. How sensitive is performance to imperfect instance masks?**
>
> Thank you for raising this interesting topic! To simulate imperfect instance masks, we replace the GT masks in COCO with the predicted masks from Mask R-CNN [3] (34.7 AP), which tend to have coarse boundaries. Using these masks, we apply the same copy-paste strategy to create mix-content images. Experimental results of M2A on FS L\&F and Road Anomaly are presented below, showing that while using imperfect masks leads to a slight performance drop compared to GT masks, our OOD fine-tuning method remains effective. We attribute this to the fact that the fine-tuning mainly captures the overall OOD patterns. Imperfect masks may introduce noise into the OOD instance-level prototypes and thus slightly weaken the feature separation objective. But the impact is limited, showcasing the robustness of our method.
>
> Results on FS L\&F:
> |Mask Type|sIoU|PPV|F1*|AuPRC|FPR$_{95}$|
> |-|-|-|-|-|-|
> |GT|39.16|38.10|34.80|90.54|1.68
> |Mask-RCNN|38.39|36.60|33.86|88.73|1.91
>
> Results on Road Anomaly:
> |Mask Type|sIoU|PPV|F1*|AuPRC|FPR$_{95}$|
> |-|-|-|-|-|-|
> |GT|53.70|51.46|50.97|75.70|16.31
> |Mask-RCNN|50.58|51.07|47.92|75.37|16.70
>
> **Reference**
>
> [1] Wang K, et al. Panet: Few-shot image semantic segmentation with prototype alignment. CVPR'19
>
> [2] Okazawa A. Interclass prototype relation for few-shot segmentation. ECCV'22
>
> [3] He K, et al. Mask r-cnn. ICCV'17

---

> > ### Comment · Reviewer_hxoy · 2025-08-05
> >
> > Thanks for the authors' response. I believe most of my concerns are addressed. However, I would encourage the authors to state clearly why reproduced results are used in the experiment section to enhance reproducibility. Also, the results of "imperfect instance masks" shall be presented in the paper, which would significantly enhance the proposed methods' applicability.

---

> > > ### Author Response · Authors · 2025-08-05
> > >
> > > Thank you very much for your encouraging feedback and constructive comments!
> > >
> > > To ensure a consistent experimental setup, we reproduce the results of existing OOD fine-tuning strategies (e.g., M2A). This enables a fair evaluation of how our instance-level extension improves the performance of each baseline. If we were to directly adopt the benchmark results reported in the original papers, the comparison would be less accurate due to minor discrepancies between their setups (such as hardware discrepancy and random seed discrepancy, as we report the mean of 3 runs) and our reproduced baselines. Although these differences are relatively small, using our own reproduced results allows for a more precise assessment of the performance gains brought by our instance-level extension.
> > >
> > > In the revised version, we will clearly clarify the rationale by reporting the reproduced results in the experiment section. For better reproducibility, we will also make our source code publicly available. Moreover, we will incorporate the results regarding “imperfect instance masks” in response to Q1 in the revised paper. We strongly believe that your insightful suggestions will greatly contribute to improving the overall quality of our revised paper.
> > >
> > > We sincerely appreciate your time and effort in carefully reviewing our paper once again.

---

### Official Review · Reviewer_nMfG · 2025-07-01

**Clarity:** 4
**Significance:** 3
**Originality:** 3
**Rating:** 5
**Confidence:** 3

**Summary:**

This paper presents a fine-tuning strategy for instance-level segmentation in images with small anomalies. The strategy is flexible, and the authors apply it to three segmentation approaches. Along with extensive benchmarks and experiments, the authors provide a theoretical analysis of why prior approaches fail at this task. Finally, some qualitative results are also provided.

**Questions:**

1. Does $L_{pred}$ contain $L_{in}$ twice? It contains $L_{glob}$ which contains $L_{in}$ and $L_{ins}$ which also contains $L_{in}$. I am concerned about the relative weight of the $L_{in}$ ($\alpha$ becomes pretty meaningless) in the optimization. Is this a problem? Did you notice any increased focused on the in distribution pixels due to this if so?

2. "the selection of these metrics and datasets aligns with official guidelines". I think the datasets and metrics seem appropriate, but what official guidelines are you referring to? Being more explicit about this may strengthen your paper and choice of these metrics and datasets.

3. Are the ablation results consistent on the other datasets? Why was the SMIYC-RA chosen for the ablation?

4. Does focusing on improving segmentation on small anomalies hurt segmentation on large anomalies? The quantative results don't seem to clearly distinguish between them.

**Ethical Concerns:**

["NO or VERY MINOR ethics concerns only"]

**Final Justification:**

During the rebuttal period, the authors addressed each weakness presented in my initial review. The authors included updated tables and descriptions of figures to be included in the paper that address my main initial concerns of qualitative results, missing confidence bounds, and limited ablation. I believe this is a solid conference paper that will be interesting to the NeurIPS and anomaly segmentation communities. Therefore, I recommend acceptance.

**Limitations:**

The limitations are kept as a short paragraph in the appendix. Perhaps they should be moved to the main text of the paper.

No potential negative societal impacts are discussed. As the paper aims to improve image segmentation for OOD scenarios, these may be worth mentioning even if they are not unique to this paper.

**Quality:**

4

**Strengths And Weaknesses:**

## Strengths

- The proposed approach has strong quantitative results across multiple datasets and against numerous baselines.
- The approach can be applied on top of existing methods to improve segmentation.
- The paper provides a theoretical analysis of why prior approaches may not work well.
- The paper is well written and detailed.

## Weaknesses

- The qualitative results are not convincing. For example, consider Figure 4. It is unclear what is ID or OOD in these images. In row 2, the small traffic cones are small anomalies. Is the large one also an anomaly? In the top row, the LNOIB method has some yellow artifact to the right of the anomaly. Does this consistently appear in images segmented using this method? Maybe demonstrating the qualitative performance on different images from the other datasets could convince the readers that the approach practically works well.

- The authors presented means of 3 random seeds as the results, but the variance or confidence bounds are not included.

- Please see my questions for more weaknesses and concerns

## Quality

The paper is well written and the figures supplement the text well. Overall, the paper is high quality.

## Clarity

The paper is detailed and the appendix is extensive. The approach is relatively simple and seems clear when considering other approaches in this sub-field.

## Significance

According to the results presented in the paper, the approach seems to improve small anomaly segmentation. This is likely significant for many CV applications. However, some of the results seem to be minor improvements with no confidence intervals provided, so it is hard to know whether there is a statistical or practical improvement in some cases (specifically in pixel-level scenarios, component-level seems more obvious).

## Originality

The fine-tuning strategy and theorems presented seem novel.

---

> ### Author Rebuttal · Authors · 2025-07-27
>
> We sincerely appreciate your encouraging feedback and insightful suggestions. We will give a detailed response to each of your constructive comments below.
>
> **W1-1.The qualitative results are not convincing. For example, consider Figure 4. It is unclear what is ID or OOD in these images. In row 2, the small traffic cones are small anomalies. Is the large one also an anomaly?**
>
> Thank you for pointing out this issue! In the revised version, we will include the ground-truth annotations (ID vs. OOD) for each image shown in Figure 4 to enhance clarity. Specifically, in row 2, all the traffic cones—including the large one—are considered anomalies. This is because anomalies are defined as objects that fall outside the semantic categories of Cityscapes. Therefore, traffic cones are treated as anomalies in the evaluation datasets.
>
> **W1-2. In the top row, the LNOIB method has some yellow artifact to the right of the anomaly. Does this consistently appear in images segmented using this method?**
>
> Thank you for your sharp observation! After identifying this issue, we evaluate the fine-tuned M2A model with two additional random seeds (as reported in the main results) and observe no noisy information near the anomaly in this image. We also conduct OOD fine-tuning with our method using two other seeds, and the predictions for this anomaly remain clean. Additionally, we use the original M2A model from Figure 4 to visualize results on approximately 50 images across five datasets. Most anomalies are accurately segmented, with only about 2 anomalies (each covering less than 0.5% of the image) showing relatively noisy information around the boundaries. We suspect this is due to the downsampling and upsampling processes in the network, which may randomly affect a few small anomalies. While this has minimal impact on overall performance—since most small anomalies are well handled—it highlights an interesting direction for future work.
>
> In the revised paper, we will provide an improved illustration generated using other seeds in Figure 4. Additionally, we will include these minority imperfect cases and give a discussion of the potential reasons in the Limitations and Future Work section to highlight potential avenues for further research.
>
> **W1-3. Maybe demonstrating the qualitative performance on different images from the other datasets could convince the readers that the approach practically works well.**
>
> Thank you for your insightful suggestion! We have used the M2A model to visualize results on approximately 50 images across five datasets. The results show that, with OOD fine-tuning using our method, most small anomalies are well segmented, significantly outperforming previous global-level strategies. As image uploads are not permitted during the rebuttal phase, we will include additional visualization results from FS L\&F and Road Anomaly datasets in Figure 4 of the revised paper to better illustrate this improvement.
>
> **W2. The authors presented means of 3 random seeds as the results, but the variance or confidence bounds are not included.**
>
> Thank you for highlighting this important point! We present the mean and standard deviation of our main results across three runs below.
>
> Results on component-level metrics:
> | |  | SMIYC-RA| | |SMIYC-RO| | |FS-Static| | |FS L\&F| | |Road Anomaly| |
> |-|:-:|:-:|:-:|:-:|:-:|:-:|:-:|:-:|:-:|:-:|:-:|:-:|:-:|:-:|:-:|
> |**Method**|**sIoU**|**PPV**|**F1***|**sIoU**|**PPV**|**F1***|**sIoU**|**PPV**|**F1***|**sIoU**|**PPV**|**F1***|**sIoU**|**PPV**|**F1***|
> |EM+LNOIB|58.73|47.42|43.90|50.86|58.80|48.57|44.38|37.20|37.79|34.74|29.56|23.97|35.72|33.18|27.30|
> | |(±0.26)|(±0.21)|(±0.35)|(±0.38)|(±0.22)|(±0.09)|(±0.30)|(±0.41)|(±0.25)|(±0.17)|(±0.24)|(±0.10)|(±0.18)|(±0.05)|(±0.29)
> |PEBAL+LNOIB|52.17|39.55|33.48|32.88|12.90|13.46|38.84|29.75|28.44|27.36|35.78|17.73|37.68|30.91|32.65|
> | |(±0.41)|(±0.13)|(±0.44)|(±0.30)|(±0.06)|(±0.16)|(±0.25)|(±0.19)|(±0.09)|(±0.26)|(±0.33)|(±0.14)|(±0.33)|(±0.15)|(±0.12)
> |M2A+LNOIB|63.15|57.37|60.08|61.50|73.71|70.25|52.08|44.93|42.96|39.16|38.10|34.80|53.70|51.46|50.97|
> | |(±0.48)|(±0.53)|(±0.27)|(±0.07)|(±0.51)|(±0.46)|(±0.18)|(±0.20)|(±0.30)|(±0.31)|(±0.17)|(±0.19)|(±0.38)|(±0.25)|(±0.08)
>
> Results on pixel-level metrics:
> | |SMIYC-RA| |SMIYC-RO| |FS-Static| |FS L\&F| |Road Anomaly| |
> |-|:-:|:-:|:-:|:-:|:-:|:-:|:-:|:-:|:-:|:-:|
> |**Method**|**AuPRC**|**FPR$_{95}$**|**AuPRC**|**FPR$_{95}$**|**AuPRC**|**FPR$_{95}$**|**AuPRC**|**FPR$_{95}$**|**AuPRC**|**FPR$_{95}$**|
> | EM+LNOIB|86.26|15.28|84.47|2.39|86.95|8.89|81.68|5.18|72.42|19.74
> | |(±0.47)|(±0.11)|(±0.35)|(±0.06)|(±0.21)|(±0.10)|(±0.24)|(±0.07)|(±0.68)|(±0.17)
> | PEBAL+LNOIB|61.46|22.03|31.30|18.44|83.81|4.47|64.50|3.82|65.35|22.77
> | |(±0.44)|(±0.18)|(±0.07)|(±0.14)|(±0.52)|(±0.03)|(±0.40)|(±0.11)|(±0.14)|(±0.10)
> | M2A+LNOIB|92.20|11.34|92.58|0.33|75.19|8.27|90.54|1.68|79.38|13.29
> | |(±0.11)|(±0.07)|(±0.42)|(±0.04)|(±0.55)|(±0.11)|(±0.58)|(±0.05)|(±0.22)|(±0.08)
>
> As shown in the tables above, our method demonstrates stability across different seeds, with standard deviations typically below 1\%. We will add these results to Tables 1 and 2 in the revised paper.
>
> **Q1. Does $L_{pred}$ contain $L_{in}$ twice? It contains $L_{glob}$ which contains $L_{in}$ and $L_{ins}$ which also contains $L_{in}$. I am concerned about the relative weight of the $L_{in}$ ($\alpha$ becomes pretty meaningless) in the optimization. Is this a problem? Did you notice any increased focused on the in distribution pixels due to this if so?**
>
> Yes, although $L_{pred}$ contains $L_{in}$ twice, this does not pose a problem. When expanding $L_{pred}$ in Eq. (6), we find that the coefficient of $L_{in}$ is 1, while the coefficients of the second terms in $L_{glob}$ (Eq. (1)) and $L_{ins}$ (Eq. (5)) — denoted as $L_{out}$ and $L_{outins}$ — are $\alpha$ and $(1-\alpha)$, respectively.
>
> Therefore, the ID loss $L_{in}$ remains independent of $\alpha$, which solely controls the balance between the global-level OOD loss $L_{out}$ and the instance-level OOD loss $L_{outins}$. Moreover, the weight of $L_{in}$ is 1, and the weight of the combined OOD losses ($L_{out}+L_{outins}$) is also equal to 1 ($\alpha+1-\alpha$), ensuring that no excessive emphasis is placed on the ID regions. This balanced formulation keeps the contrastive objective of ID region and OOD region within the same range, enabling effective optimization for both ID and OOD regions.
>
> **Q2. Being more explicit about "the selection of these metrics and datasets aligns with official guidelines".**
>
> Thank you for pointing out this issue! Under your kind reminder, we will use "official metrics" instead of "official guidelines" in the revised paper to better convey our intention. The datasets we use adopt official pixel-level metrics to evaluate overall performance on OOD regions, and component-level metrics to assess average accuracy across all anomalies. These official metrics have been commonly used in prior works. In contrast, the instance-level metric is a more recent addition, which we find helpful in validating our main claim on segmenting small anomalies.
>
> **Q3. Are the ablation results consistent on the other datasets? Why was the SMIYC-RA chosen for the ablation?**
>
> Thank you for your constructive concern! The ablation results are consistent on other datasets such as FS L\&F and Road Anomaly. The reason we chose SMIYC-RA for the ablation is that we think one dataset is okay and we choose the commonly used one in prior works.
>
> Motivated by your helpful concern, we conduct more ablation studies with M2A on FS L\&F and Road Anomaly datasets, including the combination of $L_{glob}$ and $L_{ins}$ in prediction-based objective (please refer to the tables below), the combination of $L_{sem}$ and $L_{near}$ in feature separation objective (please refer to our response to Reviewer vbkr, Q3-3), as well as the hyperparameters of $\tau$ and $M$ in in feature separation objective (please refer to our response to Reviewer qk6n, W2).
>
> Results on FS L\&F:
> | $L_{glob}$ | $L_{ins}$ | sIoU | PPV | F1* | AuPRC | FPR$_{95}$ |
> |-|-|-|-|-|-|-|
> |✓| |26.51|19.43|23.76|89.91|1.85
> | |✓|25.18|15.69|17.22|57.30|8.97
> |✓|✓|**37.58**|**35.10**|**34.23**|**90.38**|**1.77**
>
> Results on Road Anomaly:
> |$L_{glob}$|$L_{ins}$|sIoU|PPV|F1*|AuPRC|FPR$_{95}$|
> |-|-|-|-|-|-|-|
> |✓| |47.43|40.80|44.57|75.70|16.31
> | |✓|35.80|31.68|37.39|53.04|21.54
> |✓|✓|**48.99**|**50.14**|**46.73**|**78.01**|**14.27**
>
> Based on the results obtained on the FS L&F and Road Anomaly datasets, our approach demonstrates robust ablation performances across different datasets.
>
> **Q4. Does focusing on improving segmentation on small anomalies hurt segmentation on large anomalies? The quantative results don't seem to clearly distinguish between them.**
>
> Thank you for your insightful concern! We would like to clarify that the OOD fine-tuning with our proposed method does not compromise the segmentation performance on large anomalies. As evidenced in Table 2, the pixel-level metrics even improve when using LNOIB. Given that large anomalies contribute substantially to pixel-level scores, any degradation in their segmentation would have led to a noticeable drop in overall performance. The observed improvements thus indicate that segmentation on large anomalies is well preserved.
>
> **L1. The limitations are kept as a short paragraph in the appendix. Perhaps they should be moved to the main text of the paper.**
>
> Thank you for your great suggestion! We will move the limitations to the main text of our paper in the revised version.
>
> **L2. No potential negative societal impacts are discussed.**
>
> Thank you for your helpful comment! As to the potential negative societal impact, while our method shows significant improvements over prior approaches in autonomous driving scenarios, it still cannot guarantee 100% accuracy. Therefore, there remains a potential risk when deploying it in safety-critical applications. We will add this point to the revised paper.

---

> > ### Comment · Reviewer_nMfG · 2025-08-04
> >
> > Thank you for your detailed response. I believe my concerns will be properly addressed in the revised version of the paper. Therefore, I will maintain my high rating.

---

> > > ### Author Response · Authors · 2025-08-04
> > >
> > > Thank you so much for your recognition of our work and your constructive comment, which is very helpful in improving the quality of our work.

---

### Official Review · Reviewer_Fsxc · 2025-07-03

**Clarity:** 4
**Significance:** 3
**Originality:** 3
**Rating:** 5
**Confidence:** 3

**Summary:**

This paper aims to improve instance level anomaly segmentation (AS) by reconsidering common loss formulations. The paper observes that the commonly used global / pixel prediction objective function de-emphasizes small anomalies. Straightforward theoretical justifications are made to support their observations. Based on this observation a novel instance wise loss is added that is averaged over the number of pixels in the object and is therefore size independent providing an emphasis on detecting each instance. Next, the paper proposes prototype feature matching to emphasize the importance of small instances. The final objective is proposed with 4 subcomponents that are all weighted with 5 hyperparameters. This construction is generalizable across specific forms of loss functions. The paper then provides experimental support for their model showing improved performance on instance and component wise metrics while maintaining and sometimes improving pixel wise metrics. An ablation is also provided to weigh the contributions of each of the proposed losses.

**Questions:**

- What hyperparameter selection strategy or specific values should be considered for each of the loss weights? How likely is this to hinder adoption by the wider community?

**Ethical Concerns:**

["NO or VERY MINOR ethics concerns only"]

**Final Justification:**

The author response provided additional guidelines for setting the several additional hyperparameters. I maintain my rating as the paper provides a method motivated by theoretical findings that will be interesting to the NeurIPS community.

**Limitations:**

The authors address limitations in the supplemental materials.

**Quality:**

4

**Strengths And Weaknesses:**

Strengths:
- The paper follows a clear logical flow of arguments and the writing is clear
- The theoretical support provided is clear and supports the arguments made in the paper
- Experiments support the aims set out in the construction of the loss

Weaknesses:
- There are several hyperparameters necessary for weighing each of the components of the loss. While a basic search has been provided over the set of parameters, this is an added complexity in future adoption of this multi-objective loss.

*Note*: I do not closely follow the field of anomaly segmentation and so may not have identified missing reference to related anomaly segmentation methods and am not an expert in the datasets and benchmarks commonly used in this field.

---

> ### Author Rebuttal · Authors · 2025-07-25
>
> We sincerely appreciate your encouraging feedback and valuable suggestion. We will give a detailed response to your insightful question below.
>
> **W1&Q1. What hyperparameter selection strategy or specific values should be considered for each of the loss weights? How likely is this to hinder adoption by the wider community?**
>
> Thank you for raising this interesting and insightful topic! We think that although these hyperparameters may increase the complexity in future adoption to other segmentation tasks, they also provide different selections to suit the corresponding tasks. We will give an analysis of the selection recommendation of each hyperparameter below.
>
> 1) Balancing Factor $\alpha$ (Global vs. Instance-Level Objective):
> The parameter $\alpha$ controls the balance between the global-level and instance-level objectives. In anomaly segmentation, both large and small anomalies are critical for ensuring safety, so we set $\alpha = 0.5$ by default. For tasks less sensitive to small objects, a larger value (e.g., $\alpha = 0.75$) may be more appropriate. Conversely, for applications that prioritize accurate segmentation of small objects, a smaller value (e.g., $\alpha = 0.25$) is preferable. Thus, the choice of $\alpha$ should be guided by the task's sensitivity to object scale.
>
> 2) Balancing Factor $\beta$ ($L_{sem}$ vs. $L_{near}$):
> The hyperparameter $\beta$ adjusts the trade-off between the ID semantic loss ($L_{sem}$) and the nearest-neighbor loss ($L_{near}$). Both losses serve complementary roles: $L_{sem}$ encourages global separation of OOD prototype, while $L_{near}$ ensures local separation of OOD features. We recommend starting with $\beta = 0.5$ as a balanced setting. If further tuning is needed, binary search around values like $\beta = 0.3$ or $\beta = 0.7$ can be used to identify task-specific optima.
>
> 3) Weight $\gamma_2$ (Feature Separation Objective):
> The parameter $\gamma_2$ controls the contribution of the feature separation objective (with $\gamma_1 = 1$ fixed). As the prediction-based objective directly adopts OOD masks for optimization, this objective serves as an auxiliary mechanism to enhance the separability of OOD features in the representation space. A very large $\gamma_2$ (e.g., $\geq 10$) may overly emphasize this component and is generally not recommended. Conversely, very small values (e.g., $\leq 0.01$) may render it ineffective. We suggest starting with $\gamma_2=0.1$ and $\gamma_2=1$, and using binary search for fine-grained explorations.
>
> 4) Threshold $\tau$ (Prototype Filtering):
> The threshold $\tau$ filters pixel-level features to build ID category prototypes. A larger $\tau$ yields purer but less informative prototypes, while a smaller $\tau$ retains more feature variation at the cost of increased noise. We recommend setting $\tau \in [0.5, 0.9]$ depending on task requirements. For tasks that value prototype richness and can tolerate some noise, $\tau = 0.5$ is a reasonable starting point. For applications demanding high purity, initial value like $\tau = 0.7$ is more suitable. Further refinement can be conducted using binary search.
>
> 5) Number of Neighbors $M$ (in Nearest Neighbor Loss):
> The parameter $M$ determines how many ID prototypes are considered when computing the nearest neighbor loss. This loss encourages OOD prototypes to stay distant from the closest ID prototypes. Setting $M$ too large may dilute this effect. We recommend starting with $M = 1$ and incrementally increasing it up to $M = 5$ to find the best value. This small range is generally sufficient for most tasks.
>
> Once again, we sincerely appreciate your insightful suggestion! It has encouraged us to consider the broader applicability and adaptability of our method across diverse segmentation tasks.

---

> > ### Comment · Reviewer_Fsxc · 2025-08-07
> >
> > Thank you for your response. The paper provides a well motivated approach based on theoretical analysis. The provided suggestions for hyperparameter settings is helpful however it remains a concern. I maintain my original rating.

---

> > > ### Author Response · Authors · 2025-08-07
> > >
> > > Thank you so much for your recognition of our paper and your insightful comments! We truly believe that adopting our instance-level paradigm to other domains would be a very interesting and promising future work.

---

### Official Review · Reviewer_vbkr · 2025-07-03

**Clarity:** 3
**Significance:** 2
**Originality:** 3
**Rating:** 4
**Confidence:** 3

**Summary:**

The paper introduces a novel framework called LNOIB aimed at improving the detection of out-of-distribution (OOD) instances in anomaly segmentation tasks. It addresses the limitation of existing OOD fine-tuning strategies that often fail to effectively segment small anomalies, proposing an instance-level fine-tuning approach to enhance the detection of both large and small anomalies. The framework is supported by a comprehensive theoretical analysis, which strengthens the soundness of the proposed method.

**Questions:**

1.Figure 4 is not sufficiently clear. Could the authors provide the ground-truth annotations for the images to facilitate a better understanding of the results? Additionally, it remains unclear whether the introduction of LNOIB leads to an increase in false positives—i.e., misclassifying in-distribution instances as out-of-distribution. More detailed analysis or supporting data is needed to address this concern.

2.While the authors claim that LNOIB enhances the detection of small anomalies, the evidence presented is mainly visual. To substantiate this claim, quantitative metrics—such as performance breakdown by anomaly scale—should be provided. This would offer a more rigorous and comprehensive evaluation.

3.The authors mention that solely adopting L_sem may cause the OOD prototype to align with the prototype of a specific ID category. Could the authors provide specific visual examples to illustrate this point? Additionally, could the authors provide more information about the datasets used, such as the number of ID categories (K) and the similarity between different ID representations? Providing detailed metrics on the performance drop when using L_sem and L_near alone on different datasets would further support the paper’s arguments.

**Ethical Concerns:**

["NO or VERY MINOR ethics concerns only"]

**Final Justification:**

Most of my earlier concerns were addressed through clear theoretical analysis and strong experimental results. However, the core idea—instance-level loss and feature separation—has been explored in prior work in other domains, limiting novelty. Balancing these factors, I maintain my borderline accept recommendation.

**Limitations:**

yes

**Quality:**

3

**Strengths And Weaknesses:**

Strengths:

1.The paper is technically solid, offering a well-founded theoretical analysis and comprehensive experiments that aim to address key limitations of current out-of-distribution (OOD) strategies in anomaly segmentation.

2.The manuscript is clearly written and well-structured, making complex concepts easier to understand and follow.

Weaknesses:

1.Despite providing detailed theoretical analysis, the instance-level loss proposed in the paper is not entirely novel. Similar instance-level losses have been used in other detection tasks, which limits the originality of this contribution.

2.The authors claim that their feature separation objective significantly improves the detection of small anomalies, but these claims are not well-supported by experimental results or detailed analysis. While feature separation has been shown to improve model performance in other studies (e.g., [1]), the specific impact of the feature separation objective in this work remains insufficiently demonstrated.

[1]Tang J, Lu H, Xu X, et al. An incremental unified framework for small defect inspection[C]//European conference on computer vision. Cham: Springer Nature Switzerland, 2024: 307-324.

---

> ### Author Rebuttal · Authors · 2025-07-26
>
> We sincerely appreciate your encouraging feedback and valuable suggestions. We will give a detailed response to each of your insightful concerns below.
>
> **W1. Despite providing detailed theoretical analysis, the instance-level loss proposed in the paper is not entirely novel. Similar instance-level losses have been used in other detection tasks, which limits the originality of this contribution.**
>
> Thank you for your insightful comment! Emphasizing small instances is intuitive when they are critical to the task. However, our proposed instance-level objective is grounded in a rigorous theoretical foundation, which sets it apart. Specifically, our analysis reveals that existing global-level objectives tend to converge toward large anomalies. Building on this insight, our instance-level objective introduces instance-specific adaptation as a general extension of global-level objectives, rather than being tied to any particular loss function, which is unique in the field of anomaly segmentation. Moreover, incorporating this objective leads to significantly better performance on small anomalies, aligning well with our theoretical findings. Therefore, the theoretical foundation (unique to anomaly segmentation), the objective formulation (not tied to a specific loss), and the application scope are fundamentally different from those in other detection tasks.
>
> Following your helpful comment, we will include an additional subsection in Related Work to more clearly highlight the originality of our approach.
>
> **W2. The specific impact of the feature separation objective in this work remains insufficiently demonstrated.**
>
> Thank you for your constructive comment! We have gained valuable insights from paper [1], which helped us improve the experimental design for evaluating the feature separation objective. Specifically:
>
> 1) We will provide t-sne visualization to demonstrate that using only $L_{sem}$ may lead to OOD prototypes aligning with the prototype of the "road" category (please refer to our response to Q3-1 for text description);
>
> 2) We include more detailed information about the datasets, such as the number of ID categories and the similarity between some different ID semantic prototypes (please refer to our response to Q3-2);
>
> 3) We report the individual performance of $L_{sem}$ and $L_{near}$ on the FS L\&F and Road Anomaly datasets (please refer to our response to Q3-3 for experimental results).
>
> 4) Besides your valuable advice, we also provide extra theoretical analysis of why using the OOD prototype-based feature separation objective improves segmenting small anomalies (please refer to our response to Reviewer hxoy, W3).
>
> These analyses will be added in the revised version of our paper, and we will cite the reference you kindly provided, which has greatly inspired our experimental design and presentation of the feature separation objective.
>
> **Q1-1. Provide the ground-truth annotations in Figure 4.**
>
> Thank you for your insightful suggestion! We strongly agree that including the ground-truth labels (ID vs. OOD) would greatly enhance the interpretability of the results shown in Figure 4. We will incorporate the corresponding ground-truth masks as an additional column appended to the end of each row, to provide a clearer visual comparison in the revised paper.
>
> **Q1-2. Whether LNOIB leads to an increase in false positives?**
>
> Thank you for raising this important and interesting point! As shown in Table 2, the application of LNOIB consistently reduces the FPR$_{95}$ metric in most cases. It is worth noting that this metric reflects the pixel-level false positive rate, rather than an instance-level evaluation.
>
> Obtaining reliable instance-level metrics in our setting is particularly challenging, as our task focuses on anomaly semantic segmentation, which aims to distinguish between ID and OOD regions at the pixel level. Although some prior work (e.g., [2]) proposes approximating instance-level performance for anomaly detection, extracting ID instances remains difficult. This is because ID regions can include both discrete thing classes and continuous stuff classes, whereas OOD regions are typically more spatially separated and instance-like. The ambiguity in delineating ID instances—especially for amorphous or continuous regions—makes instance-level evaluation less reliable in this context.
>
> Therefore, lower FPR$_{95}$ results in Table 2 indicate that incorporating LNOIB effectively reduces the number of false positive pixels.
>
> **Q2. Quantitative metrics—such as performance breakdown by anomaly scale—should be provided.**
>
> Thank you for this very insightful suggestion! Following COCO, we define small anomaly as smaller than 32*32 pixels. We compare the performance of M2A with global-level objective and LNOIB. The results on FS L\&F and Road Anomaly are presented below.
>
> Results on L\&F of small anomaly:
> | |sIoU|PPV|F1*|AuPRC|FPR$_{95}$
> |----------|----------|----------|----------|----------|----------|
> |M2A|17.91|14.67|19.55|65.71|4.43|
> |M2A+LNOIB|**37.40**|**33.73**|**32.05**|**81.30**|**2.51**|
>
> Results on Road Anomaly of small anomaly:
> | |sIoU|PPV|F1*|AuPRC|FPR$_{95}$|
> |----------|----------|----------|----------|----------|----------|
> |M2A|24.61|25.05|18.39|52.69|21.36
> |M2A+LNOIB|**45.84**|**41.74**|**44.50**|**73.06**|**15.33**
>
> Based on the results above, both pixel-level and component-level metrics show notable improvements on small anomalies when using LNOIB, further supporting our main claim.
>
> **Q3-1. The authors mention that solely adopting $L_{sem}$ may cause the OOD prototype to align with the prototype of a specific ID category. Could the authors provide specific visual examples to illustrate this point?**
>
> Thank you for your constructive suggestion! This is an excellent idea to further illustrate the limitations of using $L_{sem}$ alone. As we are unable to upload figures in the rebuttal phase, we try to describe our observations in detail and incorporate the analysis into the revised paper.
>
> Our observations reveal that many road anomalies tend to appear directly on the road surface. When these small anomalies share similar color or texture with the road, using only $L_{sem}$ often causes their features to be pulled into the feature space of the “road” category, with a cosine similarity exceeding 0.6. In such cases, $L_{sem}$ remains relatively small—since it is averaged across $K$ categories—but fails to effectively separate OOD features from the ID “road” class, leading to misclassification.
>
> In the revised version, we will include a t-sne figure that visualizes the pixel-wise representations of such OOD instances, the “road” category, and other ID semantic categories. This will help clearly illustrate the limitations of relying solely on $L_{sem}$ and motivate the necessity of incorporating our proposed $L_{near}$ objective.
>
> **Q3-2. Provide more information about the datasets used, such as the number of ID categories ($K$) and the similarity between different ID representations.**
>
> Thank you for your valuable comment! Our study involves a total of seven datasets: Cityscapes, MS COCO, SMIYC-RA, SMIYC-RO, FS-Static, FS L&F, and Road Anomaly.
>
> In the first stage, we train a standard semantic segmentation model on Cityscapes, which contains 19 ID categories. This leads to 19×19=361 possible ID prototype pairings, making full visualization difficult. To provide a summary, we compute the average cosine similarity between category-wise prototypes using M2A with ResNet-50, which is 0.09. While a few category pairs show relatively high similarity—such as bus vs. train (0.31), car vs. truck (0.29), and traffic light vs. traffic sign (0.26)—the vast majority have low similarity (≤ 0.1), indicating well-separated feature representations.
>
> In the second stage, we create synthetic OOD data by pasting MS COCO (with 80 semantic categories) instances onto Cityscapes images (excluding those overlapping with 19 Cityscapes categories). This forms the OOD fine-tuning phase, where the original Cityscapes content is treated as ID and the pasted MS COCO regions as OOD. After this phase, the average prototype similarity slightly increases to 0.10 when using the fine-tuned model to conduct close-set semantic segmentation on Cityscapes.
>
> In the test phase, the remaining five datasets are binary-labeled (ID vs. OOD). Since they involve only a single ID class, computing prototype similarity among ID categories is no longer meaningful.
>
> **Q3-3. Provide detailed metrics on the performance drop when using $L_{sem}$ and $L_{near}$ alone on different datasets.**
>
> Thank you for this insightful suggestion! As shown in Table 3(b), we have already reported the results with and without both objectives on the SMIYC-RA. To further validate the effectiveness of combining both objectives, we additionally present results of M2A on the FS L&F and Road Anomaly datasets below.
>
> Results on FS L\&F:
> | $L_{sem}$ | $L_{near}$ | sIoU | PPV | F1* | AuPRC | FPR$_{95}$ |
> |----------|----------|----------|----------|----------|----------|----------|
> | | |26.51|19.43|23.76|89.91|1.85
> | ✓| |28.11|22.04|24.46|89.33|1.91
> | |✓|31.58|28.40|27.81|89.74|2.03
> |✓|✓|**35.85**|**33.93**|**32.10**|**90.07**|**1.72**
>
> Results on Road Anomaly:
> |$L_{sem}$|$L_{near}$|sIoU|PPV|F1*|AuPRC|FPR$_{95}$|
> |----------|----------|----------|----------|----------|----------|----------|
> | | |47.43|40.80|44.57|75.70|16.31
> |✓| |45.39|41.17|43.08|73.36|17.21
> | |✓|47.93|44.57|45.61|76.44|16.13
> |✓|✓|**50.51**|**48.87**|**50.05**|**78.26**|**15.45**
>
> The findings above are consistent across these datasets, confirming that the joint use of both objectives is indeed beneficial. We will add these findings in the revised paper to further support our paper’s arguments.
>
> **Reference**
>
> [1] Tang J, Lu H, Xu X, et al. An incremental unified framework for small defect inspection. ECCV'24.
>
> [2] Nekrasov A, et al. Ugains: Uncertainty guided anomaly instance segmentation. GCPR'23.

---

> > ### Comment · Reviewer_vbkr · 2025-08-06
> >
> > Most of my concerns have been addressed. While I acknowledge that the authors provide thorough theoretical analysis and sufficient experimental results, the core idea—instance-level loss and feature separation—has been explored in prior work within other domains. This somewhat limits the paper's novelty. Therefore, I still maintain my borderline accept recommendation.

---

> > > ### Author Response · Authors · 2025-08-06
> > >
> > > Thank you very much for your recognition of our work and your valuable comments!
> > >
> > > We would like to clarify that our work makes the following contributions that we believe constitute meaningful novelty:
> > >
> > > 1) Theoretical Foundation Tailored to Anomaly Segmentation: Unlike prior works that adopt instance-level losses based on intuition or heuristic motivations, our instance-level objective is **rigorously derived from a theoretical analysis specific to anomaly segmentation**. Our analysis reveals that existing global-level objectives inherently bias toward large anomalies. It is based on this theoretical foundation that we are able to formulate both simple and effective objectives. These objectives are not heuristically designed but are principled and theoretically grounded, enabling targeted enhancement of small anomaly segmentation. To our best knowledge, this is the first theoretical exploration of instance-level learning in anomaly segmentation, which fundamentally distinguishes our work from related methods in other domains.
> > >
> > > 2) General and Compatible Objective Design: Our instance-level objective is not a standalone loss, but rather a **general framework** that can be seamlessly integrated into a wide range of existing OOD fine-tuning methods. In contrast, similar ideas in other domains typically focus on designing **a specific loss function tailored to their unique domains**. Our formulation, by contrast, provides **a flexible and generalizable paradigm** that extends global-level objectives to the instance level for all existing OOD fine-tuning methods in anomaly segmentation. This design is fundamentally different in both intent and formulation from prior approaches in other domains.
> > >
> > > Once again, we sincerely appreciate your thoughtful feedback and the time you have devoted to reviewing our work.

---

### Official Review · Reviewer_qk6n · 2025-07-22

**Clarity:** 2
**Significance:** 2
**Originality:** 2
**Rating:** 4
**Confidence:** 4

**Summary:**

The paper proposes LNOIB, an instance-level OOD fine-tuning framework for anomaly segmentation (AS). It addresses the limitation of existing global-level OOD fine-tuning methods (e.g., EM, PEBAL, M2A) that neglect small anomalies.

**Questions:**

see weakness

**Ethical Concerns:**

["NO or VERY MINOR ethics concerns only"]

**Final Justification:**

Thanks for the authors' response. I believe most of my concerns are addressed

**Quality:**

2

**Strengths And Weaknesses:**

Strengths:

1. Well-Motivated Problem: Addresses a critical gap—neglect of small anomalies—in OOD fine-tuning for AS.

2. Theoretical Foundation: Lemma 1 and Theorem 1 formally justify why global objectives fail for small anomalies.

3. Comprehensive Experiments: Evaluated on 5 benchmarks (SMIYC, Fishyscapes, Road Anomaly), showing consistent improvements in component-level (sIoU, F1*) and instance-level (iAP) metrics (Tables 1, 5).

4. Practical Design: No added inference cost; compatible with existing methods (EM, PEBAL, M2A).

Weakness:

1. The core idea (instance-level weighting) resembles techniques in long-tail learning (e.g., class-balanced loss). The feature separation objective borrows heavily from metric learning (e.g., prototype alignment). While positioned as a "paradigm," contributions feel incremental over prior OOD fine-tuning works.

2. No analysis of how individual hyperparameters (e.g., $M$, $\tau$) affect results beyond SMIYC-RA (Figure 7).

3. Pixel-level metrics (AuPRC, FPR95) show marginal gains (Table 2), suggesting improvements primarily target small anomalies at the expense of broader robustness.

4. Theorem 2 assumes idealized inter-class separability(\epsilon)—unrealistic for complex urban scenes. Proofs (Appendix A) are sound but lack empirical validation of assumptions.

---

> ### Author Rebuttal · Authors · 2025-07-25
>
> We sincerely appreciate your time and valuable feedback on our paper. We will give a detailed response to each of your constructive comments below.
>
> **W1. The core idea (instance-level weighting) resembles techniques in long-tail learning (e.g., class-balanced loss). The feature separation objective borrows heavily from metric learning (e.g., prototype alignment). While positioned as a "paradigm," contributions feel incremental over prior OOD fine-tuning works.**
>
> Thank you for raising this insightful topic! We will clarify the key differences between our approach and the method you mentioned.
>
> (1) Instance-level weighting resembles techniques in long-tail learning (e.g., class-balanced loss).
>
> While the intuitive concept of emphasizing rare/small classes/instances may appear similar to that of long-tail learning, our core idea differs significantly in terms of theoretical foundation, objective formulation, underlying motivation, and application scope.
>
> Firstly, our method is motivated by the theoretical findings (Lemma1, Theorem 1, and Corollary 1) that global-level objectives tend to focus more on large anomalies. These theoretical analyses are fundamentally different from long-tail learning.
>
> Secondly, our instance-level objective is formulated by the variation of existing global-level objectives, rather than a specific loss function, guaranteeing its generality in anomaly segmentation. Moreover, our method is image-specific—operating at the instance level per image—rather than dataset-wide as in long-tailed learning methods [1].
>
> Thirdly, as to underlying motivation, the methods in long-tail learning often emphasize improving performance on rare categories in multi-class classification tasks. In contrast, anomaly segmentation in our work is inherently a binary problem (ID vs. OOD).
>
> At last, for application scope, while long-tailed learning aims to rebalance rare ID classes, our approach explicitly focuses on enhancing the segmentation of small anomalies, which is a different problem with distinct challenges and goals.
>
> (2) The feature separation objective borrows heavily from metric learning (e.g., prototype alignment).
>
> The use of prototypes is conceptually related to metric learning tasks, such as few-shot segmentation (FSS). However, our feature separation objective introduces significant novelties.
>
> Firstly, we introduce instance-level prototypes for OOD regions, which is the first method to construct prototype at the instance level using anomaly masks. In contrast, FSS methods generate category-level prototypes for new ID classes.
>
> Moreover, our method formulates prototype usage as an optimization problem, with the key objective of explicitly pushing OOD prototypes away from ID semantic prototypes. In contrast, prototype alignment methods [2] in FSS directly use prototypes to match query image pixels for classification, without involving any explicit optimization objective on the prototypes.
>
> Additionally, we also provide a novel theoretical foundation of how these OOD instance-level prototypes enhances segmentation performance on small anomalies in the feature separation loss (Please see our response to Reviewer hxoy, W3), which is unique in anomaly segmentation.
>
> In summary, the prototype construction (OOD instance vs. ID class), the objective of our prototype learning strategy (optimization vs. matching), and the theoretical basis differ fundamentally from the work in metric learning.
>
> (3) While positioned as a "paradigm," contributions feel incremental over prior OOD fine-tuning works.
>
> Firstly, we provide a novel theoretical analysis of why prior OOD fine-tuning works fail to segment small anomalies, which lays the foundation of our work. Moreover, our method acts as a general framework that can be effortlessly integrated into existing OOD fine-tuning strategies. Due to its broad compatibility, the design of LNOIB is intentionally simple, ensuring it fits seamlessly into a variety of pipelines. While the method may not appear complex at first glance, it effectively fills a widening gap—improving the segmentation of small anomalies, which is a known limitation of existing works. Furthermore, integrating our simple yet effective method into existing OOD fine-tuning strategies (e.g., EM, PEBAL, and M2A) leads to notable performance gains, especially on component-level and instance-level metrics. These improvements highlight the practical value of our method in accurately detecting small anomalies—an essential factor for safety-critical applications in anomaly segmentation.
>
> Following your constructive suggestion, we will add a new subsection in Related Work to briefly introduce relevant methods you mentioned and clearly delineate the differences between our approach and existing strategies.
>
> **W2. More ablation results needed.**
>
> Thank you for pointing out this issue! In our original paper, we primarily conduct ablation studies on SMIYC-RA across multiple models. Following your helpful suggestion, we further investigate whether the ablation findings generalize to other datasets. Specifically, we use M2A to explore the optimal choices of $M$ and $\tau$ on two additional datasets: FS L&F and Road Anomaly.
>
> Following the same setup as in Figure 7, we evaluate 25 combinations of $M$ and $\tau$ for each dataset. We present the results below.
>
> sIoU Results of M2A on FS L\&F (the first row represents the values of $M$, and the first column stands for the values of $\tau$):
> | |1|2|3|4|5|
> |-|-|-|-|-|-|
> |0.5|33.14|32.85|32.07|31.45|30.78
> |0.6|34.95|34.80|34.28|33.60|33.25
> |0.7|**35.85**|35.15|34.57|34.06|33.77
> |0.8|34.96|34.64|33.73|33.06|32.39
> |0.9|31.44|31.08|30.23|29.50|28.80
>
> sIoU Results of M2A on Road Anomaly (the first row represents the values of $M$, and the first column denotes the values of $\tau$):
> | |1|2|3|4|5|
> |-|-|-|-|-|-|
> |0.5|49.43|48.81|48.17|48.02|47.56
> |0.6|50.45|50.22|49.86|48.91|48.12
> |0.7|**50.51**|50.47|50.19|49.03|48.54
> |0.8|50.18|50.03|49.49|48.31|47.27
> |0.9|48.20|47.81|46.10|45.55|45.15
>
> As shown in the tables, the best performance on both FS L&F and Road Anomaly is achieved when $M=1$ and $\tau=0.7$, which is consistent with our findings on SMIYC-RA. This suggests that the selected hyperparameters generalize well across different datasets.
>
> Moreover, we conduct further ablation studies to assess the effect of individual loss components, including $L_{sem}$ and $L_{near}$ (please see our response to
> Reviewer vbkr, Q3-3), as well as $L_{glob}$ and $L_{ins}$ (please see our response to
> Reviewer nMfG, Q3). These ablation experiments performed on FS L&F and Road Anomaly are consistent to the results on SMIYC-RA.
>
> **W3. Pixel-level metrics (AuPRC, FPR95) show marginal gains (Table 2), suggesting improvements primarily target small anomalies at the expense of broader robustness.**
>
> Thank you for your valuable comment! We think that the potential expense of broader robustness may lie in two aspects: 1) The decreased performance on large anomaly; 2) The decreased performance on close-set performance.
>
> On the one hand, our method enhances the performance on small anomalies, and this improvement does not come at the expense of performance on large anomalies—both pixel-level and component-level metrics are consistently improved, as shown in Tables 1 and 2. As large anomalies account for a significant portion of the pixel-level metrics, any performance degradation on them would lead to a sharp drop in the overall pixel-level metrics. Thus, our method maintains strong performance on large anomalies.
>
> On the other hand, as shown in Figure 3 and Appendix D, fine-tuning with our method results in only a modest drop in closed-set semantic segmentation accuracy, which is on par with previous OOD fine-tuning approaches.
>
> **W4. Theorem 2 assumes idealized inter-class separability—unrealistic for complex urban scenes. Proofs (Appendix A) are sound but lack empirical validation of assumptions.**
>
> Thank you for your insightful suggestion! Following your helpful advice, we conduct additional experiments to examine the similarity between category-wise prototypes on Cityscapes, which contains complex urban scene images. Specifically, we utilize EM (WideResNet38) and M2A (ResNet50) to compute the cosine similarity between prototypes, and the resulting average values are 0.13 and 0.09, respectively. While a few category pairs exhibit relatively high similarity—such as bus vs. train (0.31), car vs. truck (0.29), and traffic light vs. traffic sign (0.26)—the majority of category pairs show low similarity values (≤ 0.1). This indicates that, for well-trained semantic segmentation models on large-scale datasets like Cityscapes, most category-wise prototypes are fairly dissimilar. Although these category-wise prototypes are not rigorously orthogonal, this empirical observation suggests that the cosine similarity values are quite small in most cases.
>
> Moreover, Theorem 2 reveals the inherent limitation of using only $L_{sem}$ and highlights the necessity of introducing $L_{near}$. To convey this insight, we adopt an idealized assumption for theoretical illustration. For empirical validation, although the prototypes of different categories are not strictly orthogonal, those of different ID categories remain well separated in the feature space. However, in cases where two ID prototypes are highly similar, relying solely on $L_{sem}$ may cause the OOD prototype to collapse into the joint feature space of these similar ID categories. In such scenarios, $L_{sem}$ still yields a small value, failing to distinguish the OOD region effectively. Therefore, incorporating $L_{near}$ becomes crucial to explicitly enforce separation between the OOD prototype and the closely aligned ID prototypes.
>
> **Reference**
>
> [1] Cui Y, et al. Class-balanced loss based on effective number of samples. CVPR'19
>
> [2] Wang K, et al. Panet: Few-shot image semantic segmentation with prototype alignment. CVPR'19

---

> > ### Author Response · Authors · 2025-08-06
> > **Looking Forward to Your Response**
> >
> > Dear Reviewer
> >
> > We hope this message finds you well, and we apologize in advance for any inconvenience it may cause.
> >
> > Thank you very much for the time and effort you have devoted to reviewing our paper, as well as for your thoughtful and valuable comments. We have carefully considered and addressed each of your concerns in the rebuttal phase.
> >
> > As the discussion period is approaching to a close and we have not yet received your follow-up response, we would greatly appreciate it if you could kindly let us know if you have any remaining concerns, or if we can provide further clarification on any aspect of our response.
> >
> > We remain fully at your disposal and are happy to assist in any way that may help.
> >
> > Thank you once again, and with our best wishes,
> >
> > Authors

---

### Note · Authors · 2025-08-12

Dear AC,

We sincerely thank you and all reviewers for your dedication and constructive feedback!

For clarity, we will refer to Reviewers hxoy, nMfG, Fsxc, vbkr, and qk6n as R1, R2, R3, R4, and R5, respectively.

We are grateful for the recognition of our novel and thorough theories analyzing why current global-level objectives fail to segment small anomalies in anomaly segmentation (**R1-R5**). Building on this foundation, we propose LNOIB, an instance-level framework compatible with existing OOD fine-tuning methods (**R1, R2, R5**), which enhances the detection of small anomalies while preserving performance on large anomalies, as validated by extensive experiments (**R1-R5**). We also appreciate that all reviewers found our work well-motivated and clearly written.

During the rebuttal and discussion phase, we carefully addressed each reviewer's comments and clarified the remaining concerns. Based on their responses, we believe most concerns have been resolved, with most reviewers providing positive ratings and insightful suggestions (**R1-R4**). As to the remaining points: 1) R1 encouraged us to add the explanations and experiments shown in the rebuttal, and we promise that we will incorporate them in the revised paper. 2) R3 questioned adapting our method to other domains, and we provided hyperparameter choices and view this as promising future work. 3) R4 noted that the core idea has been explored in other domains. We have clarified that our theoretical basis, motivation, and objective formulation are fundamentally different from other domains.

We sincerely appreciate the reviewers' engagement during the discussion period in clarifying both addressed and remaining concerns. It is worth noting that R5 did not specify the remaining concerns and appears to maintain a likely negative rating, though we believe our rebuttal has addressed his/her initial points. For Weakness 1, R1 (per discussion), R2, and R3 agree that our method is novel, and we emphasize that our task is far from other domains. For Weakness 2, we conducted ablation studies on two additional datasets in the rebuttal, recognized by R2 and R4. For Weakness 3, our method does not sacrifice performance on large anomalies or closed-set results, recognized by R1 and R2. For Weakness 4, we provided supporting empirical evidence. We believe these clarifications fully address R5's concerns.

Thank you again for your time and professionality in managing the review process.

Best regards,

Authors

---

### Decision · Program_Chairs · 2025-09-17

**Decision:**

Accept (poster)

**Comment:**

This work addresses the problem of out-of-distribution (OOD) fine-tuning for anomaly segmentation. To improve performance, especially on small anomalies where global-level objective strategies often fail, the authors propose an instance-level OOD fine-tuning framework, LNOIB. They provide theoretical justification for why global-level strategies are insufficient and argue that instance-level approaches are more effective. In addition, they introduce a feature separation objective that explicitly constrains the representation of anomalies. Experimental results demonstrate that the proposed method achieves strong performance compared to baseline approaches.

Most reviewers are in favor of accepting this work, with others leaning toward borderline acceptance. Please revise the final version according to your rebuttal. Overall, I recommend acceptance of this work.